# Looking Beyond the Known: Towards a Data Discovery Guided Open-World Object Detection

**Anay Majee**       **Amitesh Gangrade**[*]       **Rishabh Iyer**

The University of Texas at Dallas

`firstname.lastname@utdallas.edu`

## Abstract

Open-World Object Detection (OWOD) enriches traditional object detectors by enabling continual discovery and integration of unknown objects via human guidance. However, existing OWOD approaches frequently suffer from semantic confusion between known and unknown classes, alongside catastrophic forgetting, leading to diminished unknown recall and degraded known-class accuracy. To overcome these challenges, we propose *Combinatorial Open-World Detection* (**CROWD**[2]), a unified framework reformulating unknown object discovery and adaptation as an interwoven combinatorial (set-based) data-discovery (CROWD-Discover) and representation learning (CROWD-Learn) task. CROWD-Discover strategically mines unknown instances by maximizing Submodular Conditional Gain (SCG) functions, selecting representative examples distinctly dissimilar from known objects. Subsequently, CROWD-Learn employs novel combinatorial objectives that jointly disentangle known and unknown representations while maintaining discriminative coherence among known classes, thus mitigating confusion and forgetting. Extensive evaluations on OWOD benchmarks illustrate that CROWD achieves improvements of 2.83% and 2.05% in known-class accuracy on M-OWODB and S-OWODB, respectively, and nearly $2.4\times$ unknown recall compared to leading baselines.

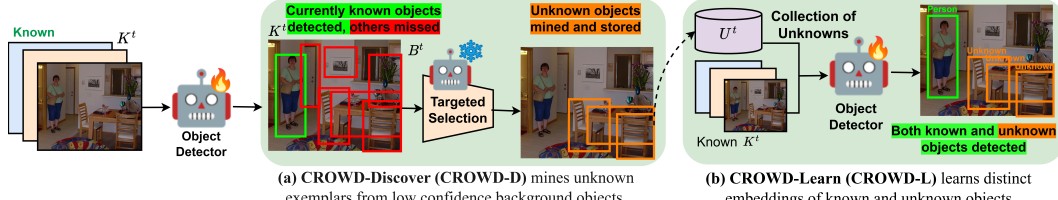

(a) **CROWD-Discover (CROWD-D)** mines unknown exemplars from low confidence background objects.

(b) **CROWD-Learn (CROWD-L)** learns distinct embeddings of known and unknown objects

Figure 1: **Overall Architecture of CROWD** showing our novel combinatorial data-discovery guided representation learning approach to (a) identify unknown objects[3] and (b) learn distinguishable representations of both known and unknown objects.

## 1 Introduction

Object Detection (OD) is central to numerous vision applications [35, 47, 30], but as shown in Figure 1 (left) conventional OD models operate under a closed-world assumption, where the object vocabulary remains fixed throughout deployment. This limitation hinders the model's ability to generalize to novel object categories. *Open-World Object Detection* (OWOD), introduced by Joseph et al. [21], addresses this by combining open-set recognition [63, 74, 65] with incremental learning [37, 12], enabling models to detect unknown objects and subsequently recognize them with minimal supervision, thus supporting continual self-improvement. However, recent efforts [64, 77, 61] reveal

---

[*]Work done as a graduate student at UTDallas.

[2]Project Page at `https://anaymajee.me/assets/project_pages/crowd.html`

39th Conference on Neural Information Processing Systems (NeurIPS 2025).

two enduring challenges: (1) confusion between known and unknown objects [21, 15, 44], and (2) catastrophic forgetting of previously learned classes [61, 64, 45]. Confusion arises due to visual similarity between unknown and known classes (e.g., truck vs. car headlamps), while forgetting stems from the lack of supervision for unknowns, causing them to be misclassified as background. Existing methods struggle to address both issues, as evidenced by low unknown recall and high Wilderness Impact scores [21] (Table 3). These limitations motivate the need for a framework that can effectively discover unknown Region-of-Interests (RoIs) and learn representations that remain distinct from known categories, thereby mitigating confusion and preserving prior knowledge.

We cast Open-World Object Detection (OWOD) as a set-based discovery and learning problem. For each task $t$, we view the *known* object classes as a collection of sets $K^t$, group all candidate *unknowns* into a single pseudo-labeled set $U^t$ [21], and treat everything else as *background* $B^t$. This formulation (Section 3.1) facilitates the incorporation of submodular functions into OWOD and gives rise to our **C**ombinato**r**ial **O**pen-**W**orld **D**etection (**CROWD**) framework. As illustrated in Figure 1, CROWD tackles OWOD as an interleaved process of data discovery (CROWD-D) and representation learning (CROWD-L), directly targeting the dual challenges of confusion and forgetting.

Starting from a OD trained only on known instances $K^t$ (Figure 1[3]), CROWD-Discover (CROWD-D) identifies representative unknowns, formulated as a combinatorial targeted selection problem. Specifically, CROWD-D maximizes the Submodular Conditional Gain (SCG, Section 3.2) between $K^t$ and candidate subsets, encouraging dissimilarity with known and background objects. Authors in [34] provide theoretical guarantees that greedy maximization [52] of SCG results in selection of samples in $U^t$ which are *dissimilar* to $K^t$ as well as $B^t$, constituting informative pseudo-labeled unknowns.

CROWD-Learn (CROWD-L) then fine-tunes the OD model on both known and mined unknowns as shown in Figure 1(b) via a novel combinatorial joint objective (Equation (1)), rooted in two families of submodular functions: SCG [34] and Total Submodular Information [11] (SIM). Maximizing SCG increases diversity between known and unknown objects, reducing feature overlap and confusion, as corroborated by Table 5. Conversely, minimizing SIM encourages intra-class cohesion within each known object, preserving discriminative features and mitigating forgetting. This formulation closely follows the observation in [48, 49] that submodular functions model cooperation [20] and diversity [38] when minimized and maximized respectively. Finally, we instantiate a family of submodular-based loss functions within CROWD-L that jointly reduce confusion and forgetting, achieving notable gains in unknown recall and known-class accuracy (Table 5). We validate our approach on two standard OWOD benchmarks, M-OWOD [21] and S-OWOD [15], demonstrating its effectiveness across diverse open-world settings. Our main contributions are -

- CROWD introduces a novel combinatorial viewpoint in OWOD by **modeling the identification of unknown instances of a given task as a data discovery problem** (CROWD-D), selecting unknown RoIs which maximize the SCG between and the known object instances.

- CROWD also introduces a novel **set-based learning paradigm CROWD-L, based on SCG functions which minimizes the cluster overlap between embeddings of known and unknown objects** while retaining the discriminative feature information from the known ones.

- Finally, CROWD demonstrates $\sim$2.4$\times$ increase in unknown recall per task alongside up to 2.8% improvement on M-OWODB and 2.1% improvement on S-OWODB in known class performance (measured as mAP) over several existing OWOD baselines.

## 2   Related Work

**Open-World Object Detection** (OWOD) first introduced in Joseph et al. [21] augmented a Faster R-CNN [58] model with contrastive clustering and an Energy-Based Unknown Classifier relying on a objectness threshold based pseudo labeling strategy. Subsequent work such as OW-DETR [15] adapted deformable DETR [76] and proposed an attention-based pseudo-labeling scheme that identifies high-activation regions as unknowns without requiring extra supervision. Further, CAT [43] improves transformer-based models by decoupling localization and classification, while introducing dual pseudo-labeling strategies, namely - model-driven and input-driven—to robustly mine unknowns. PROB [77] advanced the state of the art by modeling objectness probabilistically in the embedding

---

[3]Figure 1(a) shows a subset of background and unknown RoIs for clarity. The total number of RoIs in the original experiment is set to 512 (as in [61]) while the number of mined unknowns is set to 10 (per image).

space using a Gaussian likelihood, allowing better separation of unknowns from background without explicit negative examples. Other notable works include 2B-OCD [68], which integrates a localization-based objectness head [28], OCPL [73] enforces class-prototype separation to reduce known-unknown confusion, and UC-OWOD [69] employs feature-space regularization to suppress background misclassification. Some recent methods leverage external supervision, e.g., MViTs [46], or multimodal cues such as text for class-agnostic detection, these often fall outside strict OWOD assumptions but highlight promising directions for future research. Complementary to these, recent approaches such as RandBox [64] sidesteps detection bias via random bounding box sampling and dynamic-k filtering, while OrthogonalDet [61] enforces angular decorrelation in object features to disentangle objectness and class semantics. Interestingly, Randbox and OrthogonalDet outperforms larger models like OW-DETR, UC-OWOD etc. while using a simpler Faster-RCNN [58] based architecture. Despite substantial progress, OWOD methods continue to grapple with confusion between known and unknown objects and catastrophic forgetting during incremental adaptation recently evidenced in Xi et al. [70], motivating the development of our CROWD framework.In general our work is also related to standard object detection while CROWD-Discover (Section 3.3.1) is related to combinatorial subset selection, the related work for which is provided in Appendix A.2.

# 3 Method

## 3.1 Problem Definition: OWOD

We largely adopt the problem formulation of OWOD from Joseph et al. [21] with modifications towards a combinatorial (set-based) formulation. Given an incoming task $T_t$ where $t \in [1, n]$, an object detector $h^t(.; \theta)$ recognizes a set of known classes $K^t = \{K_1^t, K_2^t, \ldots, K_{C^k}^t\}, |K^t| = C^k$ while also accounting for unknown classes $U^t$ that may appear during inference (classes in $U^t$ are not labeled during training). Here, $K_i^t, i \in [1, C^k]$ indicates examples for each known class in $T_t$. The dataset $D^t = \{(x_i^t, y_i^t)\}_{i=1}^M$ for each task $T_t$, where each label $y_i^t$ contains $K$ object instances ($K$ can very for each image) defined by bounding box parameters $y_k^t = [c_k, x_k, y_k, w_k, h_k]$, with $c_k \in [1, C^k]$ representing the class label. The object detection model $h^t(.; \theta)$ is trained to learn newly introduced instances from labeled examples in $K^t$ while identifying unknown objects $U^t$ by assigning them a placeholder label (0). Examples in $U^t$ can be reviewed by a human expert who identifies $C^u$ new classes, allowing the model to update incrementally and produce $h^{t+1}(.; \theta)$ without retraining on the entire dataset. If $\hat{U}^t$ indicate the newly labeled set of unknown classes s.t. $|\hat{U}^t| = C^u$, then $K^{t+1} = K^t \cup \hat{U}^t$, enabling continual adaptation to new object categories over time.

## 3.2 Preliminaries: Submodularity

Adopting a set-based formulation allows us to explore combinatorial functions for OWOD. In particular we explore *Submodular functions* which are set functions exhibiting a unique diminishing returns property. Formally, a function $f : 2^{\mathcal{V}} \to \mathbb{R}$ defined on a ground set $\mathcal{V}$ is submodular if for any subset $A_i, A_j \subseteq \mathcal{V}$, it holds that $f(A_i) + f(A_j) \geq f(A_i \cup A_j) + f(A_i \cap A_j)$ [11]. These functions have been widely studied for applications such as data subset selection [27, 34, 19], active learning [67, 33, 2, 25], and video summarization [24, 26]. Typically, these tasks involve formulating subset selection or summarization as submodular maximization subject to a knapsack constraint. A classic result guarantees a $(1 - e^{-1})$ approximation factor [53] using a greedy algorithm, which can be implemented more efficiently via improved greedy strategies [53, 52]. Within this framework, **Submodular Information Functions** (SIMs) [18, 17, 3], such as Facility-Location or Graph-Cut, promote diversity when maximizing $f(A)$. On the other hand, **Submodular Conditional Gain** (SCG) [17], $H_f(A_i \mid A_j)$, captures elements in $A_i$ most dissimilar to $A_j$. Extrapolating this, Kothawade et al. [32] defines discovery of unseen, rare examples as a targeted selection problem. Further works [48, 49] have demonstrated the utility of these combinatorial functions in continuous optimization. Majee et al. [48] introduces losses inspired by SIMs to enforce intra-group compactness (when minimized) and inter-group separation (when maximized), while [49] uses SMI-based losses to account for interactions between abundant and rare samples in few-shot learning. *Motivated by these insights, CROWD proposes a novel data-discovery guided representation learning framework to dynamically identify and incrementally adapt to unknown objects.*

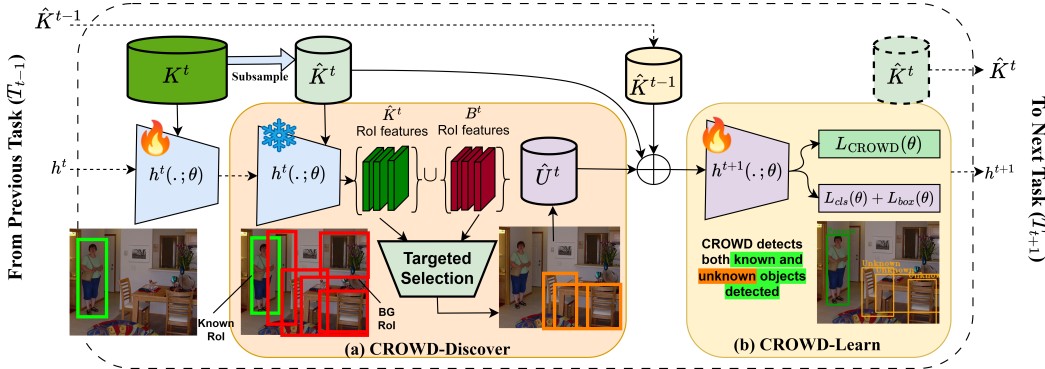

Figure 2: **Interleaved Data-Discovery and Representation Learning in CROWD** on an incoming task $T_t$. CROWD takes as input the model weights from $T_{t-1}$ and a small replay buffer of previously known classes $\hat{K}^{t-1}$, applies (a) CROWD-Learn to discover unknown RoIs and (b) CROWD-L to learn discriminative features of both known and unknown instances to return an updated model $h^{t+1}$ and the current task replay buffer $\hat{K}^t$.

### 3.3 The CROWD Framework

The problem formulation in Section 3.1 surfaces two unique challenges in the domain of OWOD - **(1)** How to *identify instances of unlabeled unknown objects* $U^t$ given labeled examples of only known ones in $K^t$ ? **(2)** How to effectively learn representations of currently known objects *without forgetting the previously known* (classes introduced in $T_i$, where $i < t$) ones ?

To this end, we introduce **C**ombinato**r**ial **O**pen-**W**orld **D**etection (**CROWD**) framework, which models the OWOD task as a interleaved set-based data discovery [32] and representation learning [48] problem. CROWD achieves this in two stages - namely **CROWD-Discover** (a.k.a. CROWD-D) and **CROWD-Learn** (a.k.a. CROWD-L) as shown in Figure 2. Given an incoming task $T_t$ we first train $h^t(.;\theta)$ on currently known classes in $D^t$. At this point, CROWD-D utilizes the frozen model weights of $h^t$ and uses a small replay buffer (typically containing examples from both previously known and currently introduced objects) to *discover* highly representative proposals of unknown classes $U^t$. We elucidate this in

---

**Algorithm 1** Discovering Unknown RoIs in CROWD-D

**Require:** A task $t$, set of RoI feature vectors $\mathtt{R} \in \mathbb{R}^{N \times d}$, Objectness scores $\mathbb{S}(.) \in \mathbb{R}^N$, Task specific Labels $y_{K^t}$, budget $\mathtt{k}$

1: */** Identify and Exclude outliers **/*
2: $\mathtt{R} \leftarrow \{r \in \mathtt{R}|\mathbb{S}(r) \geq \tau_e\}$
3: $K^t \leftarrow \texttt{HUNGARIAN-MATCHING}(\mathtt{R}, y_{K^t})$  ▷ *Known class RoI features*
4: $\mathcal{V} \leftarrow \mathtt{R} \setminus K^t$
5: */** Select Background Samples **/*
6: $B^t \leftarrow \underset{\substack{B^t \subseteq \mathcal{V} \\ |B^t| \leq \tau_b\%|\mathcal{V}|}}{\arg\max} \ H_f(B^t \mid K^t)$  ▷ *Large feature separation from $K^t$*
7: */** Select Unknown samples from $\mathtt{R} \setminus K^t$ **/*
8: $U^t \leftarrow \underset{U^t \subseteq \mathcal{V}, |U^t| \leq \mathtt{k}}{\arg\max} \ H_f(U^t \mid K^t \cup B^t)$ ▷ *Unknowns are different from $K^t \cup B^t$*
9: **return** $U^t$

---

Section 3.3.1. Subsequently, CROWD-L introduces a novel combinatorial learning strategy to rapidly finetune $h^t$ on this replay buffer (we adopt the predefined buffer in Joseph et al. [21]) to distinguish between known classes $K^t$ and unknown $U^t$ while preserving distinguishable features from the previously known classes. We discuss this in detail in Section 3.3.2.

#### 3.3.1 CROWD-Discover

During training of $h^t(.;\theta)$, label information is available only for the currently known classes $K^t$ with no labels of the previously known $K^{t-1}$ and the unknown classes $U^t$. CROWD-D tackles the challenge of **identifying potentially unknown instances** from Region-of-Interest (RoI) proposals produced by the Region-Proposal-Network (RPN) in $h^t(.;\theta)$. Unlike existing OWOD methods employ pseudo labeling [21], feature orthogonalization [61] etc. rely on the objectness score (probability of an RoI proposal to contain a foreground object), whereas CROWD-D achieves this by **modeling this task as a combinatorial data discovery problem** [32]. *Given a set of RoI proposals*

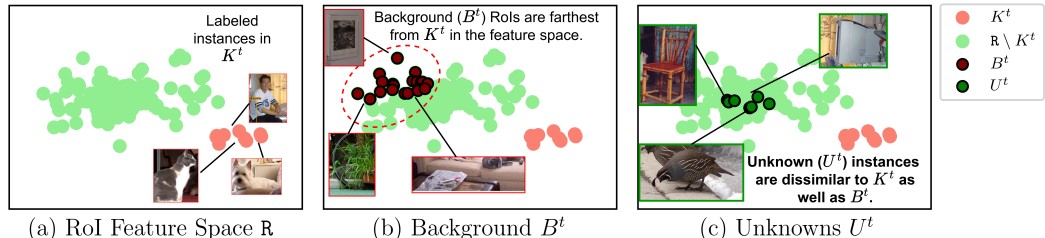

(a) RoI Feature Space R    (b) Background $B^t$    (c) Unknowns $U^t$

Figure 3: **Illustration of the data-discovery pipeline in CROWD-D** on a synthetic dataset with $|R| = 500$ and budget $k = 10$ and the underlying submodular function as Graph-Cut. CROWD-D selects $U^t$ which are both dissimilar to background $B^t$ and known $K^t$ instances.

R *and a submodular function $f$ we define the data discovery task (Algorithm 1) as a targeted selection problem which selects a set of unknown instances $U^t$ from $\mathcal{V} = R \setminus K^t$ that maximizes the SCG $H_f$ given a query set comprising of known $K^t$ and the background $B^t$ instances (line 8 in Algorithm 1).*

Here, $k$ denotes a budget for the number of unknown samples mined per image. From the definition of SCG in Section 3.2 **selected examples in $U^t$ are largely dissimilar to examples in $K^t \cup B^t$** indicating that they are neither background objects nor visually similar to known objects as shown in Figure 3(c). Interestingly, the number of known RoIs $K^t$ in R are significantly fewer than the background RoIs $B^t$ (typically $|R| = 500$ (total number of RoIs from the RPN) whereas $|K^t| \sim 10$ in MS-COCO [40] which leaves $|B^t| \sim 490$) in most OD models. To minimize the computation costs while selecting $U^t$ from this large RoI pool we exclude all RoIs with low objectness scores $\mathbb{S} < \tau_e$ (line 2 in Algorithm 1) and likely background objects $B^t$ predicted with high confidence (line 6 of Algorithm 1). Instances in $B^t$ are selected which maximize the SCG between themselves and known RoIs $K^t$ under a budget constraint of $\tau_b\%|\mathcal{V}|$ (samples that are significantly different from $K^t$). Known instances are identified by following the hungarian matching technique applied in Wang et al. [64] as shown in line 3 of Algorithm 1. The exclusion thresholds $\tau_e$ and $\tau_b$ are empirically determined to be 0.2 and 30% respectively and the underlying submodular function in our experiments is chosen to be Graph-Cut which has been evidenced in Kothawade et al. [34] to model both representation and diversity among selected examples.

### 3.3.2 CROWD-Learn

Including unknown examples $U^t$ can potentially inject noisy labels into the training data with detrimental effects. We show in Table 5 that CROWD-D alone does not handle the knowledge retention from previously known classes $K^{t-1}$, despite significant improvements on unknown class recall, causing forgetting. CROWD-L overcomes the aforementioned challenges by introducing a novel **Combinatorial representation learning** strategy inspired from recent developments [48, 49], that ensures orthogonality (separation) between embeddings in $K^t$ and $U^t$ while minimizing the effect of forgetting of $\hat{K}^{t-1}$. Here, $\hat{K}^{t-1}$ denotes a replay buffer of previously known classes.

$$
\begin{aligned}
L_{\text{CROWD}}^{self}(\theta) &= \sum_{i=1}^{C^t} f(K_i^t; \theta) \; ; \\
L_{\text{CROWD}}^{cross}(\theta) &= \sum_{i=1}^{C^t} H_f(K_i^t | U^t; \theta) = \sum_{i=1}^{C^t} f(K_i^t \cup U^t) - f(U^t) \\
L_{\text{CROWD}}(\theta) &= L_{\text{CROWD}}^{self}(\theta) - \eta L_{\text{CROWD}}^{cross}(\theta)
\end{aligned}
\tag{1}
$$

*Given a set of known $K^t \cup \hat{K}^{t-1}$, unknown $U^t$ classes alongside a submodular function $f$ we define a learning objective $L_{CROWD}(\theta)$ as shown in Equation (1) which jointly minimizes the Submodular Total Information ($L_{CROWD}^{self}$) over each known class $K_i^t \in \{K^t \cup \hat{K}^{t-1}\}$ and the SCG ($H_f$ as defined in Section 3.2) between known class $K_i^t$ and the unknown set $U^t$ ($L_{CROWD}^{cross}$). Note that CROWD-L is applied during training of task $T_t$ as a finetuning step as shown in Figure 2(b).*

Note that $f$ relies on the pairwise interaction between examples in a batch which we represent using cosine similarity $s_{ku}(\theta) = \frac{h^t(x_k,\theta)^\mathsf{T} \cdot h^t(x_u,\theta)}{||h^t(x_k,\theta)|| \cdot ||h^t(x_u,\theta)||}$ and can be different from the one used in CROWD-D decided through ablations in Section 4.2. Our loss formulation in $L_{\text{CROWD}}$ follows the observation in [48] which entails that submodular functions model cooperation [20] and diversity [38] when

Table 1: **Summary of various instantiations of CROWD-L** by varying the submodular function $f$ in $L_{\text{CROWD}}^{cross}$ and $L_{\text{CROWD}}^{self}$. Here, $\mathcal{T}$ denotes a batch with instances from $K^t \cup U^t$.

| Objective Name | Instances of $L_{\text{CROWD}}^{cross}$ | Instances of $L_{\text{CROWD}}^{self}$ |
|---|---|---|
| CROWD-GC | $\sum_{i=1}^{C^t} \frac{1}{|\mathcal{T}|}[f(K_i^t;\theta) - 2\lambda\nu \sum_{k\in K_i^t, u\in U_i^t} s_{ku}(\theta)]$ | $\sum_{i=1}^{C^t} \frac{1}{|K_i^t|}[\sum_{i\in K_i^t}\sum_{j\in\mathcal{T}\setminus U^t} s_{ij}(\theta) - \lambda\sum_{i,j\in K_i^t} s_{ij}(\theta)]$ |
| CROWD-FL | $\sum_{i=1}^{C^t} \frac{1}{|\mathcal{T}|} \sum_{n\in\mathcal{T}} \max(\max_{k\in K_i^t} s_{nk}(\theta) - \nu \max_{u\in U^t} s_{nu}(\theta), 0)$ | $\sum_{i=1}^{C^t} \frac{1}{|K_i^t|} \sum_{i\in\mathcal{T}\setminus K_i^t} \max_{j\in K_i^t} s_{ij}(\theta)$ |
| CROWD-LogDet | $\sum_{i=1}^{C^t} \frac{1}{|\mathcal{T}|} \log\det(s_{K_i^t}(\theta) - \nu^2 s_{K_i^t,U^t}(\theta) s_{U^t}^{-1}(\theta) s_{K_i^t,U^t}(\theta)^T)$ | $\sum_{i=1}^{C^t} \frac{1}{|K_i^t|} \log\det(s_{K_i^t}(\theta) + \lambda\mathbb{I}_{|K_i^t|})$ |

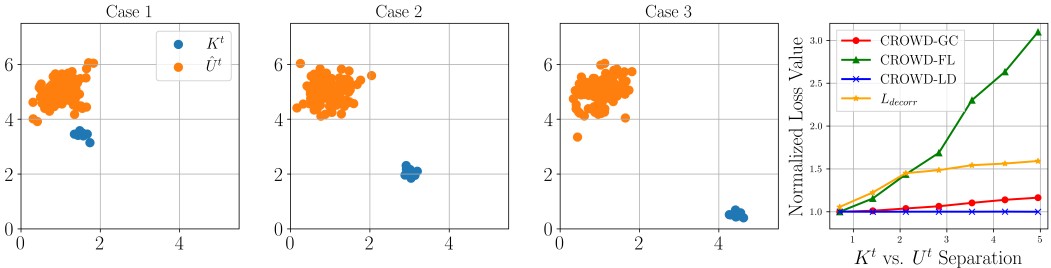

Figure 4: **Characterization of losses in CROWD-L** ($L_{\text{CROWD}}^{cross}$) on a synthetic two-cluster imbalanced dataset by increasing known vs. unknown class separation (cases 1 through 3) similar to the RoI embedding space of $h^t(.;\theta)$. The synthetic dataset generation is performed under the same seed. Here, $L_{\text{CROWD}}^{cross}$ is maximized as in Equation (1).

minimized and maximized respectively. By varying the choice of $f$ between popular submodular functions - Facility-Location (FL), Graph-Cut (GC) and Log-Determinant (LogDet) we introduce a family of loss functions summarized in Table 1 and derivations in Appendix A.3. $L_{\text{CROWD}}$ is applied to the classification head of $h^t(.;\theta)$ model during all training stages described in Sec. 4. Our novel formulation entails some interesting properties -

**(1) $L_{\textbf{CROWD}}^{self}$ retains informative known class features** : Following the insights in Jegelka and Bilmes [20] $L_{\text{CROWD}}^{self}$ which minimizes the total information contained in $K_i^t$ encourages *intra-class compactness retaining the most discriminative features from the known classes alleviating forgetting* [49].

**(2) $L_{\textbf{CROWD}}^{cross}$ models known vs. unknown separation** : As shown in Equation (1) $L_{\text{CROWD}}^{cross}$ minimizes the SCG ($H_f$) between the embeddings of the known and unknown classes. As observed in Kothawade et al. [34] maximizing SCG models *dissimilarity* between two sets. In OWOD, $L_{\text{CROWD}}^{cross}$ *promotes a large inter-class boundary between $K_i^t$ and $U^t$, minimizing their cluster overlap resulting in reduced confusion*. Further, a hyper-parameter $\eta$ controls the trade-off between intra-class compactness modeled by $L_{\text{CROWD}}^{self}$ and the inter-class separation in $L_{\text{CROWD}}$.

**(3) Sensitivity to unknown classes with varying $f$** : Table 1 highlights the instances of $L_{\text{CROWD}}$ by varying $f$. Such variations injects domain specific properties into CROWD-L critical for the OWOD tasks. As depicted in Figure 4 for $L_{\text{CROWD}}^{cross}$, under varying known vs. unknown class separation, CROWD-FL explicitly models *representation* [34] by adopting the FL based submodular function while CROWD-LogDet injects *diversity* by modeling cluster volume through Log-Determinant [18]. CROWD-GC models both representation and diversity but is not resilient to imbalance between $K^t$ and $U^t$ as CROWD-FL [48]. Thus, CROWD-FL emerges to be a suitable choice for OWOD modeling both representation and resilience to imbalance.

**(4) Generalization to Incremental Object Detection** (IOD) : As observed in several related works [21, 15, 61, 77], learning from unknown pseudo labels in OWOD benefits Incremental Object Detection (IOD) tasks. However, its important to note that CROWD relies on mined unknowns (from CROWD-D) while no unknown objects are provided in the IOD setting. This requires us to slightly modify our learning objective $L_{\text{CROWD}}^{cross}$ to model dissimilarity between currently known $K^t$ and the previously known objects $\hat{K}^{t-1}$ (a replay buffer as in Joseph et al. [21]) instead of unknowns, s.t. $L_{\text{CROWD}}^{cross}(\theta) = -\sum_{i=1}^{C^t} H_f(K_i^t|\hat{K}^{t-1};\theta)$.

Table 2: **Open-world object detection results across incremental tasks.** U-Recall and mAP (%) are reported for various baselines on M-OWOD and S-OWOD benchmarks. Best results are in **bold**.

| Method | Task 1 | | Task 2 | | | | Task 3 | | | | Task 4 | | |
|---|---|---|---|---|---|---|---|---|---|---|---|---|---|
| | U-Recall | mAP Curr. | U-Recall | mAP Prev. | Curr. | Both | U-Recall | mAP Prev. | Curr. | Both | mAP Prev. | Curr. | Both |
| *M-OWOD Benchmark Results* | | | | | | | | | | | | | |
| ORE [21] | 4.9 | 56.0 | 2.9 | 52.7 | 26.0 | 39.4 | 3.9 | 38.2 | 12.7 | 29.7 | 29.6 | 12.4 | 25.3 |
| OST [72] | - | 56.2 | - | 53.4 | 26.5 | 39.9 | - | 38.0 | 12.8 | 29.6 | 30.1 | 13.3 | 25.9 |
| OW-DETR [15] | 7.5 | 59.2 | 6.2 | 53.6 | 33.5 | 42.9 | 5.7 | 38.3 | 15.8 | 30.8 | 31.4 | 17.1 | 27.8 |
| UC-OWOD [69] | - | 50.7 | - | 33.1 | 30.5 | 31.8 | - | 28.8 | 16.3 | 24.6 | 25.6 | 15.9 | 23.2 |
| ALLOW [45] | 13.6 | 59.3 | 10.0 | 53.2 | 34.0 | 45.6 | 14.3 | 42.6 | 26.7 | 38.0 | 33.5 | 21.8 | 30.6 |
| PROB [77] | 19.4 | 59.5 | 17.4 | 55.7 | 32.2 | 44.0 | 19.6 | 43.0 | 22.2 | 36.0 | 35.7 | 18.9 | 31.5 |
| CAT [44] | 23.7 | 60.0 | 19.1 | 55.5 | 32.7 | 44.1 | 24.4 | 42.8 | 18.7 | 34.8 | 34.4 | 16.6 | 29.9 |
| RandBox [64] | 10.6 | **61.8** | 6.3 | - | - | 45.3 | 7.8 | - | - | 39.4 | - | - | 35.4 |
| OrthogonalDet [61] | 24.6 | 61.3 | 26.3 | 55.5 | 38.5 | 47.0 | 29.1 | 46.7 | 30.6 | 41.3 | 42.4 | 24.3 | 37.9 |
| **CROWD (ours)** | **57.9** | 61.7 | **53.6** | 56.7 | 38.9 | 47.8 | 69.6 | 48.0 | 31.4 | 42.5 | 42.9 | 25.4 | 38.5 |
| *S-OWOD Benchmark Results* | | | | | | | | | | | | | |
| ORE [21] | 1.5 | 61.4 | 3.9 | 56.5 | 26.1 | 40.6 | 3.6 | 38.7 | 23.7 | 33.7 | 33.6 | 26.3 | 31.8 |
| OW-DETR [15] | 5.7 | 71.5 | 6.2 | 62.8 | 27.5 | 43.8 | 6.9 | 45.2 | 24.9 | 38.5 | 38.2 | 28.1 | 33.1 |
| PROB [77] | 17.6 | 73.4 | 22.3 | 66.3 | 36.0 | 50.4 | 24.8 | 47.8 | 30.4 | 42.0 | 42.6 | 31.7 | 39.9 |
| CAT [44] | 24.0 | **74.2** | 23.0 | **67.6** | 35.5 | 50.7 | 24.6 | 51.2 | 32.6 | 45.0 | 45.4 | 35.1 | 42.8 |
| OrthogonalDet [61] | 24.6 | 71.6 | 27.9 | 64.0 | 39.9 | 51.3 | 31.9 | 52.1 | 42.2 | 48.8 | 48.7 | 38.8 | 46.2 |
| **CROWD (ours)** | **50.5** | 73.5 | **41.7** | 64.9 | 41.2 | 53.1 | 49.6 | 54.7 | 42.1 | 48.4 | 49.8 | 43.0 | 46.4 |

Table 3: **Unknown Class Metrics on M-OWODB.** Comparison of U-Recall, WI, and A-OSE across tasks (excluding Task 4 where all classes are known $U^t = \phi$). Best results are in **bold**.

| Method | Task 1 | | | Task 2 | | | Task 3 | | |
|---|---|---|---|---|---|---|---|---|---|
| | U-Recall ($\uparrow$) | WI ($\downarrow$) | A-OSE ($\downarrow$) | U-Recall ($\uparrow$) | WI ($\downarrow$) | A-OSE ($\downarrow$) | U-Recall ($\uparrow$) | WI ($\downarrow$) | A-OSE ($\downarrow$) |
| ORE [21] | 4.9 | 0.0621 | 10459 | 2.9 | 0.0282 | 10445 | 3.9 | 0.0211 | 7990 |
| OST [72] | - | 0.0417 | 4889 | - | 0.0213 | 2546 | - | 0.0146 | 2120 |
| OW-DETR [15] | 7.5 | 0.0571 | 10240 | 6.2 | 0.0278 | 8441 | 5.7 | 0.0156 | 6803 |
| PROB [77] | 19.4 | 0.0569 | 5195 | 17.4 | 0.0344 | 6452 | 19.6 | 0.0151 | 2641 |
| RandBox [64] | 10.6 | 0.0240 | 4498 | 6.3 | 0.0078 | 1880 | 7.8 | 0.0054 | 1452 |
| OrthogonalDet [61] | 24.6 | **0.0299** | 4148 | 26.3 | **0.0099** | 1791 | 29.1 | 0.0077 | 1345 |
| **CROWD (Ours)** | **57.6** | 0.0380 | 3823 | **53.6** | 0.0101 | 1508 | **69.6** | 0.0066 | 1266 |

## 4 Experiments

**Datasets** : We evaluate our approach on two well established benchmarks - **M-OWOD** [21] and **S-OWOD**[15]. M-OWOD, (*Superclass-Mixed OWOD Benchmark*) consists of images from both MS-COCO [40] and PASCAL-VOC [10] depicting 80 classes grouped into 4 tasks (20 classes per task). On the other hand, S-OWOD (*Superclass-Separated OWOD Benchmark*) consists of images from only MS-COCO dataset. Both benchmarks split the underlying data points into four distinct (non-overlapping) tasks $T_t$, where $t \in [1, 4]$. During training on a task $T_t$ the model in provided labeled examples from $T_t$ alone while at inference the model is expected to identify objects in tasks leading up to $T_t$, s.t $t \in [1, t]$. No prior knowledge of subsequent tasks $t \in [t + 1, n]$ ($n$ refers to maximum number of tasks in an experiment) are available during training and inference on $T_t$. In contrast to M-OWODB, S-OWODB introduces a distinct separation between super-categories (eg. animals, vehicles etc.) and distributes these super-categories between tasks (each task will have examples from one or more unique super-categories).

**Experimental Setup** : Following Sun et al. [61] we adopt a Faster-RCNN [58] based model with a pretrained ResNet-50 [16] backbone. Our model is trained incrementally on 4 tasks as described above with a batch size of 12, an AdamW optimizer, a base learning rate to $2.5 \times 10^{-5}$ and weight decay of $1 \times 10^{-4}$. CROWD-D utilizes the RoI features ($|R| = 500$) to mine $k = 10$ unknown instances (determined through ablation study in Appendix A.4.1) per image. The CROWD-L loss is applied across tasks as an additional head and operates on RoI features projected to a 256-dimensional feature space. We train our model on 4 NVIDIA V100 GPUs, provide additional experimental details in Appendix A.4 and release our code at https://github.com/amajee11us/CROWD.git.

**Metrics** : We use mean average precision (mAP) to evaluate known classes, partitioned into previously seen and newly introduced categories. For unknown object class, we follow OWOD conventions [21, 15] and report unknown object recall (U-Recall), as mAP is inapplicable due to incomplete annotations. To measure confusion between known and unknown classes, we report Wilderness Impact (WI) [7] and Absolute Open-Set Error (A-OSE) [51].

Table 5: **Ablation Experiments on the M-OWOD benchmark.** We report the U-Recall and mAP (all known classes) by varying the choice of selection strategies in CROWD-D and learning objectives in CROWD-L. We show that a joint (data discovery + combinatorial loss) strategy provides the best overall performance (denoted as CROWD (joint)).

| Method | Baseline | CROWD-D | CROWD-L | Task 1 U-Recall | Task 1 mAP | Task 2 U-Recall | Task 2 mAP | Task 3 U-Recall | Task 3 mAP | Task 4 mAP |
|---|---|---|---|---|---|---|---|---|---|---|
| OrthogonalDet [61] | ✓ | | | $24.6_{\pm0.04}$ | $61.3_{\pm0.11}$ | $26.3_{\pm0.01}$ | $47.0_{\pm0.06}$ | $29.1_{\pm0.01}$ | $41.3_{\pm0.10}$ | $37.9_{\pm0.09}$ |
| CROWD-D (w/ FLCG) | ✓ | ✓ | | $50.7_{\pm0.23}$ | $60.3_{\pm0.07}$ | $52.2_{\pm0.33}$ | $45.7_{\pm0.04}$ | $60.1_{\pm0.18}$ | $40.6_{\pm0.03}$ | $38.3_{\pm0.11}$ |
| CROWD-D (w/ GCCG) | ✓ | ✓ | | $57.0_{\pm0.17}$ | $61.2_{\pm0.05}$ | $54.1_{\pm0.72}$ | $45.2_{\pm0.02}$ | $69.6_{\pm0.11}$ | $40.8_{\pm0.01}$ | $38.1_{\pm0.09}$ |
| CROWD-D (w/ LogDetCG) | ✓ | ✓ | | $56.4_{\pm0.46}$ | $61.2_{\pm0.10}$ | $54.1_{\pm0.65}$ | $44.1_{\pm0.07}$ | $69.1_{\pm0.26}$ | $39.7_{\pm0.10}$ | $37.6_{\pm0.08}$ |
| CROWD-L (w/ FLCG) | ✓ | | ✓ | $25.0_{\pm0.01}$ | $61.7_{\pm0.02}$ | $26.8_{\pm0.03}$ | $47.7_{\pm0.16}$ | $28.8_{\pm0.30}$ | $42.4_{\pm0.11}$ | $38.5_{\pm0.06}$ |
| CROWD-L (w/ GCCG) | ✓ | | ✓ | $24.3_{\pm0.03}$ | $61.3_{\pm0.12}$ | $27.1_{\pm0.10}$ | $47.4_{\pm0.26}$ | $31.0_{\pm0.44}$ | $40.2_{\pm0.11}$ | $38.2_{\pm0.10}$ |
| CROWD-L (w/ LogDetCG) | ✓ | | ✓ | $22.7_{\pm0.01}$ | $59.5_{\pm0.09}$ | $27.0_{\pm0.14}$ | $44.6_{\pm0.22}$ | $27.2_{\pm0.21}$ | $38.3_{\pm0.14}$ | $36.0_{\pm0.27}$ |
| CROWD (joint) | ✓ | ✓ | ✓ | $57.9_{\pm0.33}$ | $61.7_{\pm0.02}$ | $53.6_{\pm0.41}$ | $47.8_{\pm0.02}$ | $69.6_{\pm0.26}$ | $31.4_{\pm0.03}$ | $38.5_{\pm0.07}$ |

## 4.1 Results on Benchmark OWOD and IOD tasks

**OWOD**: We compare the performance of CROWD against several existing baselines on M-OWOD and S-OWOD benchmarks as shown in Table 2. Note, that we follow Sun et al. [61] and report our results on the same seed and compute settings for fair comparisons. CROWD surpasses the latest baseline OrthogonalDet [61] by up to 2.8% and 2.1% on M-OWOD and S-OWOD benchmarks while achieving up to 2.4× gains in U-recall. For approaches like PROB [77], CAT [44] which adopt selection strategies to mine unknowns our combinatorial approach achieves up to 8.4% (on M-OWOD) improvements. This can be attributed to the contributions of CROWD-D which mines representative unknown examples effectively increasing the coverage on such objects. Also, we observe $\sim 3\%$ increase in mAP for previously known classes indicating a reduction in forgetting. The competitive results on the currently known classes (Curr. in Table 2) indicates that $h^t(.;\theta)$ enforces a stronger decision boundary between $K^t$ and $U^t$ through $L_{\text{CROWD}}^{cross}$ while retaining performance on $K^t$ through $L_{\text{CROWD}}^{self}$. Additionally, in Table 3 we show that CROWD achieves lesser confusion over existing baselines while boosting U-Recall establishing the importance of modeling OWOD as a combinatorial data-discovery problem. This is further highlighted qualitatively in Figure 5.

**IOD**: Our novel loss formulation described in Section 3.3.2 (point 4) is applied to the finetuning stage of IOD across three popular task splits from the PASCAL-VOC [10] dataset. Note, that for IOD we do not apply CROWD-D due to absence of unknown examples. Our results summarized in Table 4 and detailed in Table 9 (Appendix) shows up to 5.9% boost in overall mAP showing *better generalization to IOD* tasks while *minimizing the impact of forgetting via stronger retention of previously known classes*, a very common pitfall in IOD.

Table 4: **Results of CROWD on PASCAL VOC for three IOD tasks** shown in terms of Prev., Curr., and overall mAP.

| 10 + 10 setting | | | |
|---|---|---|---|
| | Prev. | Curr. | mAP |
| ILOD [60] | 63.2 | 63.2 | 63.2 |
| Faster ILOD [55] | 69.8 | 54.5 | 62.1 |
| PROB [77] | 66.0 | 67.2 | 66.5 |
| OrthogonalDet [61] | 69.4 | 71.8 | 67.0 |
| **CROWD (ours)** | **73.5** | **75.1** | **72.0** |
| 15 + 5 setting | | | |
| ILOD [60] | 68.3 | 58.4 | 65.8 |
| Faster ILOD [55] | 71.6 | 56.9 | 67.9 |
| PROB [77] | 73.2 | 60.8 | 70.1 |
| OrthogonalDet [61] | 74.5 | 66.9 | 72.6 |
| **CROWD (ours)** | **76.2** | **68.9** | **74.4** |
| 19 + 1 setting | | | |
| ILOD [60] | 68.5 | 62.7 | 68.2 |
| Faster ILOD [55] | 68.9 | 61.1 | 68.5 |
| PROB [77] | 73.9 | 48.5 | 72.6 |
| OrthogonalDet [61] | 73.5 | 74.5 | 73.6 |
| **CROWD (ours)** | **74.2** | **75.3** | **74.2** |

## 4.2 Ablations

We conduct ablations on the M-OWOD benchmark to analyze the contributions of individual components of CROWD. On top of the baseline method OrthogonalDet [61] we first introduce instances of CROWD-D to assess the impact of data-discovery under a fixed budget k = 10. Next, we decouple CROWD-D and introduce our novel learning objectives in CROWD-L to assess their impact on forgetting and confusion as discussed in Section 3.3.2. For each of the above steps we ablate among instances of $f$ - Graph-Cut (GC), Log-Determinant (LogDet) and Facility-Location (FL). Finally, we combine the best performing instances from CROWD-D and CROWD-L into a *joint* formulation (referred to as CROWD (joint)) as shown in Table 5 which *achieves the best*

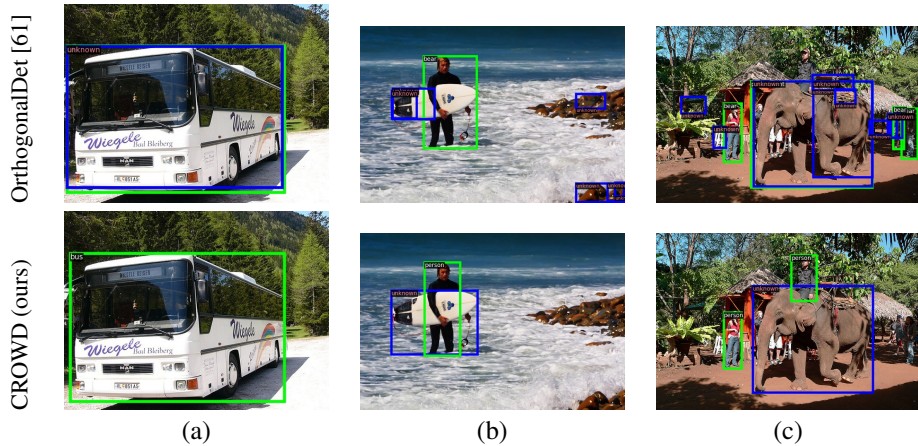

(a)                (b)                (c)

Figure 5: **Qualitative results from CROWD** contrasted against OrthogonalDet [61] showing that our approach mitigates (a) confusion (b) generalizes to unknowns and (c) reduces forgetting.

Table 6: **Ablation Experiments on the exclusion criterion $\tau_e$ and background budget $\tau_b$ in CROWD-D.** The submodular function is kept constant as Graph-Cut in CROWD-D and a CROWD-FL based learning objective is chosen in CROWD-L under constant machine seed.

| Method | Value | Task 1 | | Task 2 | | Task 3 | | Task 4 |
|---|---|---|---|---|---|---|---|---|
| | | U-Recall | mAP | U-Recall | mAP | U-Recall | mAP | mAP |
| **CROWD** ($\tau_e$) | 0.05 | 57.5 | 61.7 | 52.7 | 47.1 | 68.2 | 41.5 | 38.0 |
| $\tau_b = 30\%$ | **0.2** | 57.9 | 61.7 | 53.6 | 47.8 | 69.6 | 42.4 | 38.5 |
| | 0.5 | 53.3 | 59.5 | 49.4 | 45.9 | 65.0 | 40.1 | 37.8 |
| **CROWD** ($\tau_b$) | 10% | 57.9 | 61.7 | 53.1 | 46.8 | 69.2 | 41.3 | 37.4 |
| $\tau_e = 0.2$ | **30%** | 57.9 | 61.7 | 53.6 | 47.8 | 69.6 | 42.4 | 38.5 |
| | 50% | 55.4 | 60.0 | 50.0 | 42.1 | 63.7 | 38.9 | 34.5 |

*overall performance, balancing the tradeoff between boosting currently known class performance and retaining performance on previously learnt ones.*

**Impact of Data-Discovery in CROWD-D** : As shown in Table 5, irrespective of the choice of $f$, CROWD-D boosts the U-Recall over the baseline by introducing additional information in the form of pseudo labeled unknowns. We observe that CROWD-D (w/GCCG) ($f$ here is Graph-Cut) provides the best gains in U-Recall up to $2\times$ over the latest baseline OrthogonalDet. This follows the observation in Kothawade et al. [34] which shows that greedy maximization of GCCG models relevance (examples which are dissimilar to both $K^t$ and $U^t$) while others model diversity (CROWD-D w/LogDet) and representation (CROWD-D w/Facility-Location). Thus, we adopt GCCG based selection strategy in Algorithm 1 for our experiments in Table 2.

**Impact of k** : As stated in Section 3.3.1, k controls the number of potential unknown RoIs identified by CROWD-D per image. We ablate among several plausible values of k $\in [0, 100]$ and summarize the results in Table 10 of the Appendix. Increasing the number of identified unknowns from 0 (OrthogonalDet) to 10 shows an increase in performance of the underlying model (U-Recall) while the performance does not increase beyond 20. The increase in U-Recall can be attributed to inclusion of informative RoIs in the training loop. In fact, the mAP on known classes slightly drops below existing baselines for k $= 100$ due to inclusion of spurious background RoIs in the training pipeline.

**Impact of Combinatorial Objectives in CROWD-L** : Similar to CROWD-D we ablate on variations of $f$ to contrast between formulations summarized in Table 1. As shown in Table 5 our learning formulation, particularly CROWD-FL (based on Facility-Location) demonstrates better retention of previously known class performance while achieving competitive results on latest baseline OrthogonalDet. This follows the observation in Majee et al. [48] which demonstrates that FL based objectives model representation, retaining the most discriminative features through $L_{\text{CROWD}}^{self}$ while enforcing sufficient inter-cluster boundary between known and unknown RoI features ($L_{\text{CROWD}}^{cross}$). This also re-establishes the properties described in Figure 4 wherein CROWD-FL shows larger sensitivity to inter-cluster separation as compared to CROWD-GC, CROWD-LogDet and $L_{decorr}$ introduced in OrthogonalDet.

Table 7: **Ablation Experiments on the variation in $\eta$ in CROWD-L.** Given the Graph-Cut based selection strategy in CROWD-D we vary $\eta$ between [0.5, 1.0, 1.5] and adopt the best performing value for our pipeline in CROWD-L. The selection budget k in CROWD-D was set to 10 for all experiments and a fixed seed value.

| Method | $\eta$ | Task 1 | | Task 2 | | | | Task 3 | | | | Task 4 | | |
| | | U-Recall | mAP Curr. | U-Recall | mAP Prev. | mAP Curr. | Both | U-Recall | mAP Prev. | mAP Curr. | Both | mAP Prev. | mAP Curr. | Both |
|---|---|---|---|---|---|---|---|---|---|---|---|---|---|---|
| OrthogonalDet [61] | - | 24.6 | 61.3 | 26.3 | 55.5 | 38.5 | 47.0 | 29.1 | 46.7 | 30.6 | 41.3 | 42.4 | 24.3 | 37.9 |
| **CROWD (ours)** | 0.5 | 57.8 | 58.8 | 53.4 | 57.1 | 32.8 | 44.9 | 65.3 | 50.2 | 25.9 | 42.1 | 44.0 | 21.1 | 38.3 |
| | **1.0** | **57.9** | **61.7** | **53.6** | **56.7** | **38.9** | **47.8** | **69.6** | **48.0** | **31.4** | **42.5** | **42.9** | **25.4** | **38.5** |
| | 1.5 | 57.9 | 61.7 | 53.6 | 55.6 | 39.1 | 47.4 | 69.5 | 44.0 | 34.6 | 40.9 | 44.0 | 21.1 | 38.3 |

**Ablation on Exclusion Criterion $\tau_e$ and $\tau_b$ in CROWD-D** - At first, $\tau_e$ is an exclusion threshold which reduces the search space of CROWD-D by eliminating RoIs which have a low confidence threshold. As shown in Table 6, increasing $\tau_e$ from 0 to 1 increases performance until $\tau_e = 0.2$ and then reduces. A lower value of $\tau_e$ allows for a large search space but includes a lot of noisy background objects leading to reduced selection performance. On the other hand a large value of $\tau_e$ can potentially earmark unknown foregrounds as unknowns resulting in reduced performance.

Keeping $\tau_e$ fixed at 0.2 we ablate $\tau_b$ which controls the selection budget for backgrounds (higher the value more are the number of background RoIs identified). Increasing $\tau_b$ (percentage here) increases the fraction of RoIs treated as backgrounds. This widens the search space for the combinatorial function causing a small drop in performance due to confusions between true backgrounds and foreground unknowns. On the other hand, very large values of $\tau_b$ shrink the search space oftentimes considering unknown foregrounds as background objects showing a steep drop in performance.

**Ablation on Trade-off between $L_{\text{CROWD}}^{self}$ and $L_{\text{CROWD}}^{cross}$ in CROWD-L** - The hyper-parameter $\eta$ controls the trade-off between known-unknown class separation and known class cluster compactness discussed in Table 7. A lower value of $\eta$ does not enforce separation between currently known and unknown exemplars but enforces intra-class compactness. This results in better retention of previously known objects but a drop in currently known objects due to increased confusion with unknown exemplars. On the other hand for a large value of $\eta$ the model enforces large separation between currently known and unknown objects boosting performance on the currently knowns but suffers from catastrophic forgetting of the previously known classes.

## 5 Conclusion, Limitations and Future Work

We introduced CROWD, a novel combinatorial framework in OWOD, which reformulates OWOD as interleaved set-based discovery (CROWD-D) and representation learning (CROWD-L) tasks. Leveraging Submodular Conditional Gain (SCG) functions, CROWD-D strategically selects representative unknown instances distinctly dissimilar from known objects while CROWD-L consumes mined unknowns to preserve discriminative coherence over known classes. Our evaluations confirm that CROWD effectively addresses known vs. unknown class confusion and forgetting, achieving significant improvements in unknown recall and known-class accuracy on standard OWOD and IOD benchmarks. Despite improvements in U-Recall, operating under a fixed budget CROWD-D injects some spurious exemplars into the selected unknown pool (particularly in images with no known unknown objects). We aim to address this in future works by exploring alternative combinatorial formulations beyond SCG, and introducing stricter constraints in CROWD-D.

## Acknowledgements

We gratefully thank anonymous reviewers for their valuable comments. We would also like to extend our gratitude to our fellow researchers from the CARAML lab at UT Dallas for their suggestions. This work is supported by the National Science Foundation under Grant Numbers IIS-2106937, a gift from Google Research, an Amazon Research Award, and the Adobe Data Science Research award. Any opinions, findings, and conclusions or recommendations expressed in this material are those of the authors and do not necessarily reflect the views of the National Science Foundation, Google or Adobe.

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

# A Appendix

## A.1 Notation

Following the problem definition in the main paper we introduce the notations used in Table 8 throughout the paper.

Table 8: Collection of notations used in the paper.

| Symbol | Description |
|:---:|:---:|
| $t$ | Task identifier for each OWOD task. |
| $T_t$ | Each Task in OWOD. |
| $D^t$ | Training dataset for each task. |
| $\mathcal{T}$ | The Ground set, here refers to the mini-batch at each iteration. |
| $K^t$ | Complete set of currently known classes in $T_t$. |
| $K^{t-1}$ | Complete set of previously known classes in $T_t$. |
| $U^t$ | Complete set of unknown classes in $T_t$. |
| $\hat{K}^t$ | Predefined Replay buffer of currently known classes in $T_t$. |
| $\hat{K}^{t-1}$ | Predefined Replay buffer of previously known classes in $T_t$. |
| $h^t(x, \theta)$ | Task specific Object Detector used as feature extractor. |
| $Clf(.,.)$ | Multi-Layer Perceptron as classifier. In our case a two layer network. |
| $\theta$ | Parameters of the feature extractor. |
| $s_{A,B}(\theta)$ | Cross-Similarity between sets $A, B \in \mathcal{V}$. |
| $s_A(\theta)$ | Self-Similarity between samples in set $A \in \mathcal{T}$. |
| $f(A)$ | Submodular Information function over a set $A$. |
| $H_f(A \mid Q)$ | Submodular Conditional Gain function between sets $A$ and $Q$. |
| $L_{\text{CROWD}}(\theta)$ | Loss value computed over all known and unknown objects. |
| $L_{\text{CROWD}}^{self}(\theta)$ | Combinatorial loss computed over all known classes $K_i^t \in \mathcal{T}$. |
| $L_{\text{CROWD}}^{cross}(\theta)$ | Combinatorial loss computed between known classes $K_i^t$ and unknown classes $U^t$. |

## A.2 Additional Related Work

**Data subset selection** aims at identifying a distinct set of examples from a large pool which accurately captures the properties of the data distribution. This has rendered subset selection to be a natural choice for data-efficient machine learning tasks like Active Learning [31, 50, 59, 9], Continual Learning [29, 1], Data Summarization [31, 34] etc. Traditionally subset selection has been defined as a subsampling technique based on similarity [23], uncertainty [5] etc. or random [54]. Orthogonally, a new line of work based on combinatorial functions, particularly submodular functions [11, 18] have emerged which effectively selects informative subsets by modeling the notions of cooperation, diversity and representation [34]. These functions formulate subset selection as a greedy maximization task [52] based on several information theoretic measures like Total Information, Mutual Information, Conditional Gain etc. (discussed in Section 3.2 of the main paper). Concurrent to their success in vision [34, 31], language [39], speech [66] etc. domains,

subset selection has been used in auxiliary learning mechanisms like meta-learning [36] and data-discovery [32] targeting identification of rare or unseen examples from an unlabeled example pool. CROWD exploits this line of investigation adopting a combinatorial subset selection technique (detailed in Section 3.3.1 to discover unknown objects in the open-world setting).

**Object detection** (OD) is a fundamental task in computer vision encapsulating both localization and recognition tasks under the same roof. OD methods are traditionally grouped into two principal paradigms: single-stage and two-stage detectors. Single-stage detectors, exemplified by SSD [42], RetinaNet [41], and YOLO [57, 56], CenterNet [8] unify the processes of object localization and classification into a single feed-forward network, enabling real-time performance with relatively low computational overhead. In contrast, two-stage detectors, such as Faster R-CNN [14, 13, 58], adopt a cascaded architecture wherein a Region Proposal Network (RPN) first hypothesizes candidate object regions, followed by a refinement stage that simultaneously predicts the class and precise bounding box of each proposal. CNN based architectures struggles with the long-range dependencies, which is important for understanding the complex spatial relationships between objects at varying scales (perspective views). Transformer based models [4, 75, 6] improve upon this vulnerability by introducing a self-attention [62] mechanism based on an encoder-decoder architecture [4]. While these models achieve impressive performance in closed-world settings (all object categories present during testing are known and predefined in the training data) they under-perform in open-world scenarios when encountering unknown objects unseen during training.

**Preliminaries of Submodularity** (continued from Section 3.2) As discussed in Section 3.2 of the main paper, submodular functions have been recognized to model notions of cooperation [20], diversity [38], representation [34] and coverage [24]. Following the combinatorial formulation in Section 3.1 of the main paper we define the ground set $\mathcal{V} = \{A_1, A_2, \cdots A_N\}$, s.t. $|\mathcal{V}| = N$ and explore four different categories of submodular information functions in our work, namely -

**(1)** *Submodular Total Information* ($S_f$) which measures the total information contained in each set [11], expressed as $S_f(A_1, A_2, \ldots, A_N)$ as in Equation (2). Maximizing $S_f$ over a set $A_i$ models diversity [38] while minimizing $S_f$ models cooperation [20].

$$S_f(A_1, A_2, \ldots, A_N) = \sum_{i=1}^{N} f(A_i) \tag{2}$$

**(2)** *Submodular Conditional Gain* ($H_f$) which models the gain in information when a set $A_j$ is added to $A_i$. $H_f$ models the notion of *dissimilarity* between sets and can be expressed in Equation (3).

$$H_f(A_i|A_j) = f(A_i \cup A_j) - f(A_j), \forall i, j \in |\mathcal{V}| \tag{3}$$

Given a submodular function $f$ (can alternatively be $H_f$) tasks like selection [19, 27] and summarization [26, 24] have been modeled as a discrete optimization problem to identify a summarized set of examples $A \subseteq \mathcal{V}$ via submodular maximization under a cardinality constraint ($|A| \leq k$), i.e. $\max_{A \subseteq \mathcal{V}, |A| \leq k} f(A)$. This can be fairly approximated with a $(1 - e^{-1})$ constant factor guarantee [53] using greedy optimization techniques [52] as shown in Algorithm 2. Extending the definition of submodular functions to continuous optimization space Majee et al. [48] have proposed a set of novel family of learning objectives which minimize total information and total correlation among sets in $D_{train}$ using continuous optimization

---

**Algorithm 2** Greedy Submodular Maximization [53]

---

**Require:** Submodular function $f : 2^{\mathcal{V}} \to \mathbb{R}$, cardinality constraint k

**Ensure:** Set $A \subseteq \mathcal{V}$ maximizing $f(A)$ under cardinality constraint k

1: $A \leftarrow \emptyset$
2: **for** $j = 1$ to k **do**
3:     $e \leftarrow \arg\max_{v \in \mathcal{V} \setminus A} [f(A \cup \{v\}) - f(A)]$
4:     $A \leftarrow A \cup \{e\}$
5: **return** $A$

---

techniques like SGD. These objectives have been shown to be significantly more robust to large imbalance demonstrated in real-world tasks like longtail recognition [48] and few-shot learning [49].

### A.3 Derivations of Instances of $L_{\text{CROWD}}$

As discussed in Section 3.3.2 of the main paper, varying the choice of Submodular function $f$ in Equation (1) results in several instances of $L_{\text{CROWD}}$. Based on three popular choices of $f$ among

Facility-Location, Graph-Cut and Log-Determinant, we derive the respective formulations of $L_{\text{CROWD}}$. Note, that the derivations of $L_{\text{CROWD}}^{self}$ are adapted from Majee et al. [48] and are thus not included below.

### A.3.1 Derivation of CROWD-FL

**Theorem A.1.** *Given a set of known RoIs $K_i^t$, $i \in [1, C^t]$, a set of unknown RoIs $U^t$ ($\mathcal{T} = K^t \cup U^t$) and the Facility-Location based submodular function $f$ defined over any set $A$ s.t. $f(A) = \sum_{i \in \mathcal{T}} \max_{j \in A} s_{ij}$, we define CROWD-FL learning objective to learn the parameters $\theta$ of the model $h^t$, containing two components $L_{CROWD}^{self}$ and $L_{CROWD}^{cross}$ as shown in Equation (4). Here, $s_i j$ resembles the similarity between samples $i$ and $j$ respectively.*

$$L_{CROWD}^{self} = \sum_{i=1}^{C^t} \frac{1}{|K_i^t|} \sum_{i \in \mathcal{T} \setminus K_i^t} \max_{j \in K_i^t} s_{ij}(\theta)$$

$$L_{CROWD}^{cross}(\theta) = \sum_{i=1}^{C^t} \frac{1}{|\mathcal{T}|} \sum_{n \in \mathcal{T}} \max(\max_{k \in K_i^t} s_{nk}(\theta) - \nu \max_{u \in U^t} s_{nu}(\theta), 0) \tag{4}$$

*Proof.* From the definition of $L_{\text{CROWD}}^{cross}$ in Equation (1) we find,

$$L_{\text{CROWD}}^{cross} = \sum_{i=1}^{C^t} H_f(K_i^t | U^t)$$

$$L_{\text{CROWD}}^{cross} = \sum_{i=1}^{C^t} f(K_i^t \cup U^t) - f(U^t) \tag{5}$$

Substituting the definition of $f(A)$ over any set from the theorem in the above expression we get -

$$L_{\text{CROWD}}^{cross} = \sum_{i=1}^{C^t} H_f(K_i^t | U^t)$$

$$L_{\text{CROWD}}^{cross} = \sum_{i=1}^{C^t} \sum_{n \in \mathcal{T}} \max_{k \in K_i^t \cup U^t} s_{nk} - \sum_{n \in \mathcal{T}} \max_{u \in U^t} s_{nu}$$

$$L_{\text{CROWD}}^{cross} = \sum_{i=1}^{C^t} \sum_{n \in \mathcal{T}} \max\left(\max_{k \in K_i^t} s_{nk}, \max_{k U^t} s_{nk}\right) - \sum_{n \in \mathcal{T}} \max_{u \in U^t} s_{nu} \tag{6}$$

$$L_{\text{CROWD}}^{cross} = \sum_{i=1}^{C^t} \sum_{n \in \mathcal{T}} \max\left(\underbrace{\max_{k \in K_i^t} s_{nk}}_{\text{Term 1}} - \underbrace{\max_{u \in U^t} s_{nu}}_{\text{Term 2}}, 0\right)$$

The Term 2 in the above equation controls the degree of separation between $K_i^t$ and $U^t$. Due to this we introduce a hyper-parameter $\nu$ which we can control during model training. Sine $\nu$ is a constant it does not affect the submodular properties of $L_{\text{CROWD}}^{cross}$. The final loss formulation, normalized by the size of $\mathcal{T}$ thus becomes -

$$L_{\text{CROWD}}^{cross} = \sum_{i=1}^{C^t} \frac{1}{|\mathcal{T}|} \sum_{n \in \mathcal{T}} \max\left(\max_{k \in K_i^t} s_{nk} - \nu \max_{u \in U^t} s_{nu}, 0\right) \tag{7}$$

Additionally, we do not provide proofs for $L_{\text{CROWD}}^{self}$ since this function largely resembles the total information formulation in Majee et al. [48]. □

### A.3.2 Derivation of CROWD-GC

**Theorem A.2.** *Given a set of known RoIs $K_i^t$, $i \in [1, C^t]$, a set of unknown RoIs $U^t$ ($\mathcal{T} = K^t \cup U^t$) and the Graph-Cut based submodular function $f$ defined over any set $A$ s.t. $f(A) = \sum_{i \in \mathcal{T}} \sum_{j \in A} s_{ij} - \lambda \sum_{i,j \in A} s_{ij}$, we define CROWD-GC learning objective to learn the parameters $\theta$ of the model $h^t$ containing two components $L_{CROWD}^{self}(\theta)$ and $L_{CROWD}^{cross}(\theta)$ as shown in Equation (8). Here, $s_{ij}$ resembles the similarity between samples $i$ and $j$ respectively.*

$$L_{CROWD}^{self} = \sum_{i=1}^{C^t} \frac{1}{|K_i^t|} \Big[ \sum_{i \in K_i^t} \sum_{j \in \mathcal{T} \setminus U^t} s_{ij}(\theta) - \lambda \sum_{i,j \in K_i^t} s_{ij}(\theta) \Big]$$

$$L_{CROWD}^{cross}(\theta) = \sum_{i=1}^{C^t} \frac{1}{|\mathcal{T}|} \Big[ f(K_i^t; \theta) - 2\lambda\nu \sum_{k \in K_i^t, u \in U_i^t} s_{ku}(\theta) \Big] \tag{8}$$

*Proof.* From the definition of $L_{\text{CROWD}}^{cross}$ in Equation (1) we find,

$$L_{\text{CROWD}}^{cross} = \sum_{i=1}^{C^t} H_f(K_i^t | U^t)$$

$$L_{\text{CROWD}}^{cross} = \sum_{i=1}^{C^t} f(K_i^t \cup U^t) - f(U^t) \tag{9}$$

Substituting the definition of $f(A)$ over any set from the theorem in the above expression of $L_{\text{CROWD}}^{cross}$ we get -

$$L_{\text{CROWD}}^{cross} = \sum_{i=1}^{C^t} H_f(K_i^t | U^t) = \sum_{i=1}^{C^t} f(K_i^t \cup U^t) - f(U^t)$$

$$L_{\text{CROWD}}^{cross} = \sum_{i=1}^{C^t} \sum_{n \in \mathcal{T}} \sum_{k \in K_i^t \cup U^t} s_{nk} - \lambda \sum_{n,k \in K_i^t \cup U^t} s_{nk} - \sum_{n \in \mathcal{T}} \sum_{u \in U^t} s_{nu} + \lambda \sum_{n,u \in U^t} s_{nu}$$

$$L_{\text{CROWD}}^{cross} = \sum_{i=1}^{C^t} \sum_{n \in \mathcal{T}} \sum_{k \in K_i^t} s_{nk} + \sum_{n \in \mathcal{T}} \sum_{u \in U^t} s_{nu} - \lambda \sum_{n,k \in K_i^t \cup U^t} s_{nk} \tag{10}$$

$$- \sum_{n \in \mathcal{T}} \sum_{u \in U^t} s_{nu} + \lambda \sum_{n,u \in U^t} s_{nu}$$

The second term and the fourth term cancels out (same value with opposite signs).

$$L_{\text{CROWD}}^{cross} = \sum_{i=1}^{C^t} \sum_{n \in \mathcal{T}} \sum_{k \in K_i^t} s_{nk} - \lambda \Big( \sum_{n,k \in K_i^t \cup U^t} s_{nk} + \sum_{n,u \in U^t} s_{nu} \Big)$$

$$L_{\text{CROWD}}^{cross} = \sum_{i=1}^{C^t} \sum_{n \in \mathcal{T}} \sum_{k \in K_i^t} s_{nk} - \lambda \Big( \sum_{n,k \in K_i^t} s_{nk} + 2 \sum_{n,u \in U^t} s_{nu} \Big) \tag{11}$$

Now, rearranging the terms of the equation we get -

$$L_{\text{CROWD}}^{cross} = \sum_{i=1}^{C^t} \Big( \sum_{n \in \mathcal{T}} \sum_{k \in K_i^t} s_{nk} - \lambda \sum_{n,k \in K_i^t} s_{nk} \Big) + 2\lambda \sum_{n,u \in U^t} s_{nu}$$

$$L_{\text{CROWD}}^{cross} = \sum_{i=1}^{C^t} \underbrace{f(K_i^t)}_{\text{Term 1}} + 2\lambda \underbrace{\sum_{n,u \in U^t} s_{nu}}_{\text{Term 2}} \tag{12}$$

Similar to CROWD-FL the Term 2 in the above equation controls the degree of separation between $K_i^t$ and $U^t$. Due to this we introduce a hyper-parameter $\nu$ which we can control during model training. Sine $\nu$ is a constant it does not affect the submodular properties of $L_{\text{CROWD}}^{cross}$. The final loss formulation, normalized by the size of $\mathcal{T}$ thus becomes -

$$L_{\text{CROWD}}^{cross} = \sum_{i=1}^{C^t} f(K_i^t) + 2\lambda\nu \sum_{n,u \in U^t} s_{nu} \tag{13}$$

Additionally, we do not provide proofs for $L_{\text{CROWD}}^{self}$ since this function largely resembles the total information formulation in Majee et al. [48]. $\square$

### A.3.3 Derivation of CROWD-LogDet

**Theorem A.3.** *Given a set of known RoIs $K_i^t$, $i \in [1, C^t]$, a set of unknown RoIs $U^t$ ($\mathcal{T} = K^t \cup U^t$) and the Log-Determinant based submodular function $f$ defined over any set $A$ s.t. $f(A) = \log\det(s_A)$, we define CROWD-LogDet learning objective which contains two components $L_{CROWD}^{self}$ and $L_{CROWD}^{cross}$ as shown in Equation (14). Here, $s_ij$ resembles the similarity between samples $i$ and $j$ respectively.*

$$L_{CROWD}^{self} = \sum_{i=1}^{C^t} \frac{1}{|K_i^t|} \log\det(s_{K_i^t}(\theta) + \lambda\mathbb{I}_{|K_i^t|})$$

$$L_{CROWD}^{cross}(\theta) = \sum_{i=1}^{C^t} \frac{1}{|\mathcal{T}|} \log\det(s_{K_i^t}(\theta) - \nu^2 s_{K_i^t, U^t}(\theta) s_{U^t}^{-1}(\theta) s_{K_i^t, U^t}(\theta)^T) \tag{14}$$

*Proof.* From the definition of $L_{\text{CROWD}}^{cross}$ in Equation (1) we find,

$$L_{\text{CROWD}}^{cross} = \sum_{i=1}^{C^t} H_f(K_i^t | U^t)$$

$$L_{\text{CROWD}}^{cross} = \sum_{i=1}^{C^t} f(K_i^t \cup U^t) - f(U^t) \tag{15}$$

Substituting the definition of $f(A)$ over any set from the theorem in the above expression of $L_{\text{CROWD}}^{cross}$ we get -

$$L_{\text{CROWD}}^{cross} = \sum_{i=1}^{C^t} H_f(K_i^t | U^t) = \sum_{i=1}^{C^t} f(K_i^t \cup U^t) - f(U^t)$$

$$L_{\text{CROWD}}^{cross} = \sum_{i=1}^{C^t} \log\det(s_{K_i^t \cup U^t}) - \log\det(s_{U^t}) \tag{16}$$

$$= \sum_{i=1}^{C^t} \log \frac{\det(s_{K_i^t \cup U^t})}{\det(s_{U^t})}$$

From Schur's complement which states that given two sets A and B $\det(s_{A \cup B}) = \det(s_A).\det(s_{A \cup B} \setminus s_A)$. Replacing the term $\det(s_{K_i^t \cup U^t})$ with the above definition we get -

$$L_{\text{CROWD}}^{cross} = \sum_{i=1}^{C^t} \log \frac{\det(s_{U^t}).\det(s_{K_i^t \cup U^t} \setminus s_{U^t})}{\det(s_{U^t})}$$

$$= \sum_{i=1}^{C^t} \log\det(s_{K_i^t \cup U^t} \setminus s_{U^t}) \tag{17}$$

Table 9: **Generalization performance on Incremental Object Detection (IOD)** where we show that our CROWD approach (here only CROWD-L) when applied to the finetuning stage of IOD tasks show better generalizability. Best results are in **bold** while new classes introduced in the task are shaded gray.

| **10 + 10 setting** | aero | cycle | bird | boat | bottle | bus | car | cat | chair | cow | table | dog | horse | bike | person | plant | sheep | sofa | train | tv | mAP |
|---|---|---|---|---|---|---|---|---|---|---|---|---|---|---|---|---|---|---|---|---|---|
| ILOD [60] | 69.9 | 70.4 | 69.4 | 54.3 | 48 | 68.7 | 78.9 | 68.4 | 45.5 | 58.1 | 59.7 | 72.7 | 73.5 | 73.2 | 66.3 | 29.5 | 63.4 | 61.6 | 69.3 | 62.2 | 63.2 |
| Faster ILOD [55] | 72.8 | 75.7 | 71.2 | 60.5 | 61.7 | 70.4 | 83.3 | 76.6 | 53.1 | 72.3 | 36.7 | 70.9 | 66.8 | 67.6 | 66.1 | 24.7 | 63.1 | 48.1 | 57.1 | 43.6 | 62.1 |
| ORE [21] | 63.5 | 70.9 | 58.9 | 42.9 | 34.1 | 76.2 | 80.7 | 76.3 | 34.1 | 66.1 | 56.1 | 70.4 | 80.2 | 72.3 | 81.8 | 42.7 | 71.6 | 68.1 | 77.0 | 67.7 | 64.5 |
| Meta-ILOD [22] | 76.0 | 74.6 | 67.5 | 55.9 | 57.6 | 75.1 | 85.4 | 77.0 | 43.7 | 70.8 | 60.1 | 66.4 | 76.0 | 72.6 | 74.6 | 39.7 | 64.0 | 60.2 | 68.5 | 60.7 | 66.3 |
| ROSETTA [71] | 74.2 | 76.2 | 64.9 | 54.4 | 57.4 | 76.1 | 84.4 | 68.8 | 52.4 | 67.0 | 62.9 | 63.3 | 79.8 | 72.8 | 78.1 | 40.1 | 62.3 | 61.2 | 72.4 | 66.8 | 66.8 |
| OW-DETR[15] | 61.8 | 69.1 | 67.8 | 45.8 | 47.3 | 78.3 | 78.4 | 78.6 | 36.2 | 71.5 | 57.5 | 75.3 | 76.2 | 77.4 | 79.5 | 40.1 | 66.8 | 66.3 | 75.6 | 64.1 | 65.7 |
| PROB [77] | 70.4 | 75.4 | 67.3 | 48.1 | 55.9 | 73.5 | 78.5 | 75.4 | 42.8 | 72.2 | 64.2 | 73.8 | 76.0 | 74.8 | 75.3 | 40.2 | 66.2 | 73.3 | 64.4 | 64.0 | 66.5 |
| CAT [44] | 76.5 | 75.7 | 67.0 | 51.0 | 62.4 | 73.2 | 82.3 | 83.7 | 42.7 | 64.4 | 56.8 | 74.1 | 75.8 | 79.2 | 78.1 | 39.9 | 65.1 | 59.6 | 78.4 | 67.4 | 67.7 |
| OrthogonalDet [61][1] | 82.9 | 80.1 | 75.8 | 64.3 | 60.6 | 81.5 | 87.9 | 54.9 | 48 | 82.1 | 57.7 | 63.5 | 80.5 | 77.6 | 78.2 | 38.9 | 69.8 | 62.8 | 76.9 | 64.2 | 69.41 |
| **CROWD (ours)** | 84.1 | 84.5 | 73.9 | 60.0 | 65.1 | 80.1 | 89.3 | 82.7 | 53.3 | 77.4 | 63.4 | 78.5 | 80.9 | 83.4 | 83.9 | 46.5 | 72.6 | 60.9 | 77.9 | 71.5 | **73.5** |

| **15 + 5 setting** | aero | cycle | bird | boat | bottle | bus | car | cat | chair | cow | table | dog | horse | bike | person | plant | sheep | sofa | train | tv | mAP |
|---|---|---|---|---|---|---|---|---|---|---|---|---|---|---|---|---|---|---|---|---|---|
| ILOD [60] | 70.5 | 79.2 | 68.8 | 59.1 | 53.2 | 75.4 | 79.4 | 78.8 | 46.6 | 59.4 | 59.0 | 75.8 | 71.8 | 78.6 | 69.6 | 33.7 | 61.5 | 63.1 | 71.7 | 62.2 | 65.8 |
| Faster ILOD [55] | 66.5 | 78.1 | 71.8 | 54.6 | 61.4 | 68.4 | 82.6 | 82.7 | 52.1 | 74.3 | 63.1 | 78.6 | 80.5 | 78.4 | 80.4 | 36.7 | 61.7 | 59.3 | 67.9 | 59.1 | 67.9 |
| ORE [21] | 75.4 | 81.0 | 67.1 | 51.9 | 55.7 | 77.2 | 85.6 | 81.7 | 46.1 | 76.2 | 55.4 | 76.7 | 86.2 | 78.5 | 82.1 | 32.8 | 63.6 | 54.7 | 77.7 | 64.6 | 68.5 |
| Meta-ILOD [22] | 78.4 | 79.7 | 66.9 | 54.8 | 56.2 | 77.7 | 84.6 | 79.1 | 47.7 | 75.0 | 61.8 | 74.7 | 81.6 | 77.5 | 80.2 | 37.8 | 58.0 | 54.6 | 73.0 | 56.1 | 67.8 |
| ROSETTA [71] | 76.5 | 77.5 | 65.1 | 56.0 | 60.0 | 78.3 | 85.5 | 78.7 | 49.5 | 68.2 | 67.4 | 71.2 | 83.9 | 75.7 | 82.0 | 43.0 | 60.6 | 64.1 | 72.8 | 67.4 | 69.2 |
| OW-DETR [15] | 77.1 | 76.5 | 69.2 | 51.3 | 61.3 | 79.8 | 84.2 | 81.0 | 49.7 | 79.6 | 58.1 | 79.0 | 83.1 | 67.8 | 85.4 | 33.2 | 65.1 | 62.0 | 73.9 | 65.0 | 69.4 |
| PROB [77] | 77.9 | 77.0 | 77.5 | 56.7 | 63.9 | 75.0 | 85.5 | 82.3 | 50.0 | 78.5 | 63.1 | 75.8 | 80.0 | 78.3 | 77.2 | 38.4 | 69.8 | 57.1 | 73.7 | 64.9 | 70.1 |
| CAT [44] | 75.3 | 81.0 | 84.4 | 64.5 | 56.6 | 74.4 | 84.1 | 86.6 | 53.0 | 70.1 | 72.4 | 83.4 | 85.5 | 81.6 | 81.0 | 32.0 | 58.6 | 60.7 | 81.6 | 63.5 | 72.2 |
| OrthogonalDet [61][1] | 81.8 | 79.3 | 71.0 | 71.0 | 58.8 | 62.1 | 82.6 | 89.7 | 79.8 | 47.0 | 80.5 | 61.1 | 79.9 | 80.2 | 81.6 | 44.2 | 65.5 | 71.5 | 75.6 | 74.2 | 72.6 |
| **CROWD (ours)** | 82.8 | 80.6 | 72.5 | 59.6 | 61.3 | 83.1 | 89.3 | 83 | 49.2 | 86.1 | 62.2 | 83.7 | 86 | 80.3 | 82.8 | 46.1 | 80 | 63.7 | 79.5 | 75.6 | **74.4** |

| **19 + 1 setting** | aero | cycle | bird | boat | bottle | bus | car | cat | chair | cow | table | dog | horse | bike | person | plant | sheep | sofa | train | tv | mAP |
|---|---|---|---|---|---|---|---|---|---|---|---|---|---|---|---|---|---|---|---|---|---|
| ILOD [60] | 69.4 | 79.3 | 69.5 | 57.4 | 45.4 | 78.4 | 79.1 | 80.5 | 45.7 | 76.3 | 64.8 | 77.2 | 80.8 | 77.5 | 70.1 | 42.3 | 67.5 | 64.4 | 76.7 | 62.7 | 68.2 |
| Faster ILOD [55] | 64.2 | 74.7 | 73.2 | 55.5 | 53.7 | 70.8 | 82.9 | 82.6 | 51.6 | 79.7 | 58.7 | 78.8 | 81.8 | 75.3 | 77.4 | 43.1 | 73.8 | 61.7 | 69.8 | 61.1 | 68.5 |
| ORE [21] | 67.3 | 76.8 | 60 | 48.4 | 58.8 | 81.1 | 86.5 | 75.8 | 41.5 | 79.6 | 54.6 | 72.8 | 85.9 | 81.7 | 82.4 | 44.8 | 75.8 | 68.2 | 75.7 | 60.1 | 68.8 |
| Meta-ILOD [22] | 78.2 | 77.5 | 69.4 | 55.0 | 56.0 | 78.4 | 84.2 | 79.2 | 46.6 | 79.0 | 63.2 | 78.5 | 82.7 | 79.1 | 79.9 | 44.1 | 73.2 | 66.3 | 76.4 | 57.6 | 70.2 |
| ROSETTA [71] | 75.3 | 77.9 | 65.3 | 56.2 | 55.3 | 79.6 | 84.6 | 72.9 | 49.2 | 73.7 | 68.3 | 71.0 | 78.9 | 77.7 | 80.7 | 44.0 | 69.6 | 68.5 | 76.1 | 68.3 | 69.6 |
| OW-DETR [15] | 70.5 | 77.2 | 73.8 | 54.0 | 55.6 | 79.0 | 80.8 | 80.6 | 43.2 | 80.4 | 53.5 | 77.5 | 89.5 | 82.0 | 74.7 | 43.3 | 71.9 | 66.6 | 79.4 | 62.0 | 70.2 |
| PROB [77] | 80.3 | 78.9 | 77.6 | 59.7 | 63.7 | 75.2 | 86.0 | 83.9 | 53.7 | 82.8 | 66.5 | 82.7 | 80.6 | 83.8 | 77.9 | 48.9 | 74.5 | 69.9 | 77.6 | 48.5 | 72.6 |
| CAT [44] | 86.0 | 85.8 | 78.8 | 65.3 | 61.3 | 71.4 | 84.8 | 84.8 | 52.9 | 78.4 | 71.6 | 82.7 | 83.8 | 81.2 | 80.7 | 43.7 | 75.9 | 58.5 | 85.2 | 61.1 | 73.8 |
| OrthogonalDet [61][1] | 81.8 | 82.6 | 77.0 | 56.3 | 66.0 | 74.4 | 88.5 | 78.7 | 51.2 | 84.3 | 63.1 | 84.4 | 81.3 | 78.8 | 80.9 | 46.8 | 77.9 | 68.6 | 74.1 | 74.5 | 73.6 |
| **CROWD (ours)** | 81.7 | 80.3 | 77.4 | 57.2 | 66.8 | 80.7 | 87.1 | 67.9 | 49.4 | 87.3 | 65.6 | 84.2 | 85.4 | 79.9 | 81.6 | 48.6 | 77.0 | 69.0 | 82.2 | 75.3 | **74.2** |

Following Schur's complement yet again which states that $s_{A \cup B} \setminus s_A = s_B - s_{A,B}^T s_A^{-1} s_{A,B}$, where $s_{A,B}$ refers to the cross-similarities between sets $A$ and $B$ while $s_A$ and $s_B$ represent the corresponding self-similarities and substitute this definition into the aforementioned equation as -

$$L_{\text{CROWD}}^{cross} = \sum_{i=1}^{C^t} \log \det(s_{K_i^t} - s_{K_i^t, U^t} s_{U^t}^{-1} s_{K_i^t, U^t}^T) \tag{18}$$

Normalizing this term with the size of the ground set $|\mathcal{T}|$ and introducing the hyper-parameter $\nu$ which trades-off between inter-cluster separation and intra-cluster compactness, we derive the function for $L_{\text{CROWD}}^{cross}$ as -

$$L_{\text{CROWD}}^{cross} = \sum_{i=1}^{C^t} \frac{1}{|\mathcal{T}|} \log \det(s_{K_i^t} - \nu^2 s_{K_i^t, U^t} s_{U^t}^{-1} s_{K_i^t, U^t}^T) \tag{19}$$

Similar to previously derived objectives, we do not provide proofs for $L_{\text{CROWD}}^{self}$ since this function largely resembles the total information formulation in Majee et al. [48]. □

## A.4 Additional Experimental Details

In this section we provide additional experimental details for training our CROWD approach on M-OWOD, S-OWOD and IOD benchmarks discussed in Section 4 of the main paper.

**M-OWOD and S-OWOD benchmarks** - M-OWOD and S-OWOD benchmarks are created from MS-COCO [40] and split into 4 tasks $T_t$, where $t \in [1, 4]$ detailed in the "Datasets" section in Section 4. For each task, the model in provided labeled examples from $T_t$ alone while at inference the model is expected to identify objects in tasks leading up to $T_t$, s.t $t \in [1, t]$. We split the training

---
[1]This is a reproduction of the results from OrthogonalDet from their public repo.

Table 10: **Ablation Experiments on the variation in k in CROWD-D.** Given the Graph-Cut based selection strategy in Algorithm 1 of CROWD-D and the CROWD-FL based learning objective in CROWD-L we vary k in [0, 100] and adopt the best performing budget for our pipeline.

| Method | Budget k | Task 1 | | Task 2 | | Task 3 | | Task 4 |
|---|---|---|---|---|---|---|---|---|
| | | U-Recall | mAP | U-Recall | mAP | U-Recall | mAP | mAP |
| OrthogonalDet [61] | - | 24.6 | 61.3 | 26.3 | 47.0 | 29.1 | 41.3 | 37.9 |
| **CROWD (ours)** | 5 | 51.2 | 61.0 | 49.1 | 45.9 | 62.7 | 40.3 | 37.4 |
| | **10** | 57.9 | 61.7 | 53.6 | 47.8 | 69.6 | 42.4 | 38.5 |
| | 30 | 58.4 | 61.7 | 53.5 | 48.0 | 70.1 | 42.4 | 38.5 |
| | 100 | 57.5 | 59.3 | 53.7 | 44.3 | 70.9 | 38.8 | 32.0 |

into two splits. In the first stage the model is exposed only to the currently known classes $K^t$ and the learnt model $h^t$ biases on labeled examples in $K^t$. At the end of the first stage CROWD-D kicks in and selects representative unknowns as described in Section 3.3.1. Lets call in $U^t$. Next, we store a replay buffer of the currently known objects $\hat{K}^t$, s.t. $\hat{K}^t \subseteq K^t$. Following this, we combine $\hat{K}^t$, $K^t$ and a replay buffer from the previous task $\hat{K}^{t-1}$ into a single dataset to finetune $h^t$ using CROWD-L. As detailed in Section 3.3.2 this ensures known vs. unknown separation wile retaining discriminative features from known classes.

**IOD benchmarks** - In contrast to OWOD, IOD does not encounter unknowns during model training but experiences heavy catastrophic forgetting on previously known classes $K^{t-1}$. Following recent benchmarks like Sun et al. [61], Zohar et al. [77], Joseph et al. [21] we evaluate the IOD performance of CROWD on PASCAL-VOC benchmark on three settings produced by varying the number of newly added classes - 10 + 10, 15 + 5, 19 + 1 as shown in Table 9. In the absence of unknowns we do not apply CROWD-D and only rely on CROWD-L applied to the finetuning stage of IOD. Following latest works we adopt a replay based learning technique whichh stores a small subset of the previously known objects $\hat{K}^{t-1}$ in a buffer. $\hat{K}^{t-1}$ combined with the newly introduced classes $K^t$ is used to finetune $h^t$. This also requires us to slightly modify the formulation of $L_{\text{CROWD}}^{cross}$ as detailed in Section 3.3.2. For each setting $h^t$ is trained on a batch size of 12 for 3000 iterations using an AdamW optimizer, a base learning rate to $2.5 \times 10^{-5}$ and weight decay of $1 \times 10^{-4}$.

### A.4.1 Ablation on Selection Budget k

As detailed in Section 3.3.1, the parameter k dictates how many candidate unknown RoIs CROWD-D selects per image. We conduct an ablation over several plausible settings of k within the interval [0, 100], and present the outcome in Table 10. Fo this experiment we keep the choice of submodular function $f$ in CROWD-D as Graph-Cut and Facility-Location (CROWD-FL) for CROWD-L following the results of the ablation experiments in Table 5 in the main paper. Notably, raising k from 0 (i.e., OrthogonalDet) to 10 yields a marked uplift in the model's unknown–recall (U-Recall), yet further increases beyond k = 20 confer no additional gains. This initial boost in U-Recall stems from the integration of truly informative RoIs into the training loop. However, when k reaches its upper bound of 100, the mean average precision (mAP) on known classes experiences a slight decline relative to existing baselines—a consequence of inadvertently incorporating spurious background proposals.

### A.4.2 Results on Synthetic Datasets - CROWD-D

In addition to the illustrations provided in Figure 3 we contrast the selection performance of CROWD-D by varying the underlying submodular function $f$ between Grap-Cut (GC), Facility-Location (FL) and Log-Determinant (LogDet) on synthetic datasets as shown in Figure 6. The use of synthetic datasets provide us with complete control over the embedding space allowing us to pathologically inject imbalance, inter-cluster separation etc. in a compute efficient fashion. Particularly in our experiments we use a two-cluster imbalanced setup mimicking the RoI embedding space in Faster-RCNN [58] model. Similar to Sun et al. [61] the number of known class and unknown class feature vectors are severely imbalanced with total number of RoIs $R = 500$ and the number of knowns $|K^t| = 10$. $R$ and $K^t$ are sampled from a normal distribution with fixed variance values. The LogDet based selection strategy enforces the notion of diversity in the selection mechanism which does not select representative unknowns negatively impacting OWOD performance as shown in Table 5. The

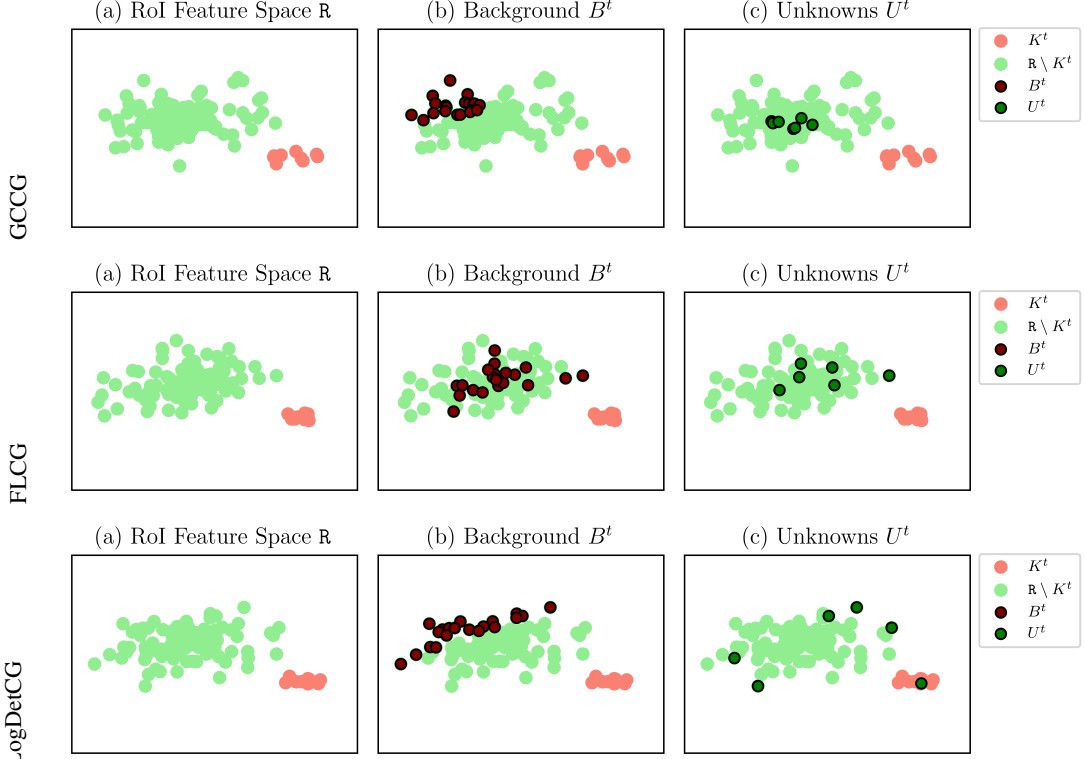

Figure 6: **CROWD-D results on synthetic dataset** contrasted against instances of popular submodular functions - Graph-Cut, Facility-Location and Log-Determinant. Graph-Cut based selection strategy models both representation and diversity resulting in the best possible choice of unknown instances in $U^t$.

FL based selection strategy models representation as shown in Figure 6 alone during selection resulting in erroneous selection of background instances negatively affecting OWOD performance. Lastly, GC based selection strategy shown in Figure 6 models notions of both diversity and representation selecting diverse backgrounds $B^t$ farthest to $K^t$ as well as representative unknowns $U^t$. This results in GC based selection strategy to produce the best overall results as shown in Table 2.

