# OpenReview forum: "Looking Beyond the Known: Towards a Data Discovery Guided Open-World Object Detection"
_NeurIPS.cc/2025/Conference — NeurIPS 2025 poster_

### Official Review · Reviewer_5m2P · 2025-06-30

**Clarity:** 2
**Significance:** 3
**Originality:** 3
**Rating:** 4
**Confidence:** 4

**Summary:**

The paper proposes a novel Open World Object Detection (OWOD) approach named CROWD, which innovatively decomposes the OWOD task into two stages: CROWD-D and CROWD-L. By maximizing the SCG function, the method sequentially selects candidate regions that significantly differ from the known class feature representations as background and unknown objects, thereby effectively enhancing the detection capability for unknown objects. Furthermore, by minimizing the clustering overlap between the embeddings of known and unknown objects, the method mitigates confusion between known and unknown categories. Overall, CROWD demonstrates substantial innovation and advancement in the recognition of unknown classes compared to prior work.

**Questions:**

See weaknesses

**Ethical Concerns:**

["NO or VERY MINOR ethics concerns only"]

**Final Justification:**

Thank you for the patient response. I maintain the positive score.

**Limitations:**

yes

**Paper Formatting Concerns:**

No formatting issues in this paper.

**Quality:**

3

**Strengths And Weaknesses:**

Strengths:
1.   This paper reformulates unknown object discovery and adaptation as an interwoven combinatorial data-discovery and representation learning task. This idea has novelty
2.   CROWD has significantly improved the recall rate of unknown objects compared to previous methods


Weaknesses ：

1.	The CROWD-D module for unknown class discovery appears to exhibit a strong dependence on the annotations of known classes. In this framework, both background and unknown object pseudo-labels are derived based on their similarity to the feature representations of known classes. Under conditions where the annotations and feature diversity of known categories are limited, the reliability of unknown candidate selection，driven by the maximization of the SCG function may be compromised. Moreover, variations in the distribution of known classes could potentially impact the robustness of the model in discovering unknown categories.

2.	The formulation and notations used in the paper could benefit from better consistency. Specifically, the connection between the SCG function introduced earlier and its usage in later sections could be clarified further. Additionally, the symbols used in the loss function formulation should be explained in more detail for better understanding.

---

> ### Author Rebuttal · Authors · 2025-07-29
>
> **[W1] “The CROWD-D module …”** - We agree with the reviewer that the selection of $U^t$ in CROWD-D is dependent on the known $K^t$. This dependence on $K^t$ has been a common trend in OWOD since its induction through ORE (even in latest SoTA paper OrthogonalDet) since examples $K^t$ are the only source of reliable labeled examples (often used as anchors). We would also like to point the reviewer to Gupta et al., 2022 (OW-DETR) who introduced a super-class separated benchmark (S-OWODB in our paper) ensuring varying distributions across tasks (a scenario in question) where CROWD demonstrates superior performance over SoTA methods. As shown by the authors of PRISM (Kothawade et al., 2022) for auxiliary tasks, submodular maximization of SCG shows exemplary performance in mining rare/unseen instances where the size of the query set is very small compared to the ground set. This is because SCG relies on gain in information in contrast to simple distance/similarity measures used in other approaches. Our work extends this property to OWOD tasks with modifications discussed in Sec. 3.3.
>
> **[W2] “The formulation …”** - We understand the concern of the reviewer. Due to lack of space in the main paper we provided the key property of SCG functions alone in Sec. 3.2 relevant to the formulation of CROWD in Sec. 3.3 and beyond. However, we would like to point the reviewer to Sec. A.2 of the Appendix alongside Sec. 3.2 (main paper) where we detail out the preliminaries of submodular functions and the mathematical modeling of SCG supporting the property of feature dissimilarity that SCG models upon submodular maximization (also detailed in Algorithm 2 of Appendix). This property is consistently used across CROWD-D and CROWD-L with differences only in the optimization strategy. In CROWD-D (a selection formulation) we maximize SCG using Greedy Optimization (Nemhauser etal., 1978) to select examples in $U^t$ dissimilar to $K^t$ while in CROWD-L (specifically in $L_{\text{CROWD}}^{\text{cross}}(\theta)$) SCG is maximized using SGD to model separation (dissimilarity) between $K^t \cup B^t$ and $U^t$ (mined in CROWD-D).
>
> Additionally, for the notations used in the loss formulation of CROWD-L we would like to point the reviewer to Table 6 in the Appendix where we explicitly call out the definitions of each term used in our formulation. As suggested by the reviewer we would be happy to include this in the main paper to improve readability.

---

### Official Review · Reviewer_bAGd · 2025-07-02

**Clarity:** 3
**Significance:** 3
**Originality:** 3
**Rating:** 4
**Confidence:** 3

**Summary:**

The paper deals with the problem of classifying known objects and detecting unknown objects. The tasks are continually discovered, and the model has to adapt to new tasks, giving rise to two challenges: catastrophic forgetting and confusion among known/unknown objects. The proposed method tackles the problem by proposing two major components. First component discovers the known and unknown objects by leveraging Submodular Conditional Gain and second components learns the representation of known/unknown objects. Experiments on benchmarks reveal effectiveness of the proposed method for the task of Open-World Object Detection.

**Questions:**

See weakness section

**Ethical Concerns:**

["NO or VERY MINOR ethics concerns only"]

**Final Justification:**

My concerns about missing studies have been addressed in rebuttal. Overall, I am positive about the paper.

**Limitations:**

YES for limitations

For societal impact, authors have given answer of [NA]. I suggest the following as the societal impact of the work.

Object detection are extensively used in real world scenario. The proposed work would be meaningful in scenarios for such real-world scenario (where setting is often open-world) and model has to continually adapt to new tasks.

**Quality:**

3

**Strengths And Weaknesses:**

**Strength**

- The paper tackles the challenging but important problem of continual object detection in open-world setting.
- The design of the proposed components is well motivated: data discovery for each category, then learning representation.
- Ablations on impact of each component illustrates the importance of the design.
- Performance is compared against substantial number of baselines.

**Weakness**

- Missing study of the impact of each component of the proposed loss function (hyperparameter η).
- The paper mentions how thresholds τe and τb are determined. Could authors show the details?

---

> ### Author Rebuttal · Authors · 2025-07-29
>
> **[W1] “Missing study …”** - We thank the reviewer for pointing this out. As stated in lines 230 - 231 $\eta$ controls the trade-off between known-unknown class separation and known class cluster compactness. A lower value of $\eta$ does not enforce separation between currently known and unknown exemplars but enforces intra-class compactness. This results in better retention of previously known objects but a drop in currently known objects due to increased confusion with unknown exemplars. On the other hand for a large value of $\eta$ the model enforces large separation between currently known and unknown objects boosting performance on the currently knowns but suffers from catastrophic forgetting of the previously known classes. This is shown in Tab. R2 below. We adopt the value of $\eta$ which provides the best overall performance (a middle ground between forgetting previously knowns and improving upon currently known classes).
>
> *Table R2: Ablation Experiments on the variation in $\eta$ in CROWD-L. Given the Graph-Cut based selection strategy in CROWD-D we vary $\eta$ between [0.5, 1.0, 1.5] and adopt the best performing value for our pipeline in CROWD-L. The selection budget $\mathtt{k}$ in CROWD-D was set to 10 for all experiments and a fixed seed value.*
>
> | **Method**                   | $\eta$ | **Task 1** U-Recall | **Task 1** mAP Curr. | **Task 2** U-Recall | **Task 2** mAP Prev. | **Task 2** mAP Curr. | **Task 2** mAP Both | **Task 3** U-Recall | **Task 3** mAP Prev. | **Task 3** mAP Curr. | **Task 3** mAP Both | **Task 4** mAP Prev. | **Task 4** mAP Curr. | **Task 4** mAP Both |
> |------------------------------|-------|---------------------|----------------------|---------------------|----------------------|----------------------|---------------------|---------------------|----------------------|----------------------|---------------------|----------------------|----------------------|---------------------|
> | OrthogonalDet [Sun etal., 24]       |   -   | 24.6                | 61.3                 | 26.3                | 55.5                 | 38.5                 | 47.0                | 29.1                | 46.7                 | 30.6                 | 41.3                | 42.4                 | 24.3                 | 37.9                |
> | **CROWD (ours)**             | 0.5   | 57.8                | 58.8                 | 53.4                | 57.1                 | 32.8                 | 44.9                | 65.3                | 50.2                 | 25.9                 | 42.1                | 44.0                 | 21.1                 | 38.3                |
> | **CROWD (ours)**             | 1.0   | 57.9            | **61.7**             | 53.6            | 56.7             | 38.9             | **47.8**            | 69.6           | 48.0             | 31.4             | **42.5**            | 42.9             | 25.4             | **38.5**            |
> | **CROWD (ours)**             | 1.5   | 57.9                | 61.7                 | 53.6                | 55.6                 | 39.1                 | 47.4                | 69.5                | 44.0                 | 34.6                 | 40.9                | 40.2                 | 28.0                 | 37.2                |
>
> **[W2] “The paper mentions …”** - Both $\tau_e$ and $\tau_b$ are determined empirically. $\tau_e$ is an exclusion threshold which reduces the search space of CROWD-D by eliminating RoIs which have a low confidence threshold. As shown in Tab R3 below, increasing $\tau_e$ from 0 to 1 increases performance until $\tau_e = 0.2$ and then reduces. A lower value of $\tau_e$ allows for a large search space but includes a lot of noisy background objects leading to reduced selection performance. On the other hand a large value of $\tau_e$ can potentially earmark unknown foregrounds as unknowns resulting in reduced performance.
>
> Next in Tab. R3 we keep $\tau_e$ fixed at 0.2 and ablate $\tau_b$ which controls the selection budget for backgrounds (higher the value more are the number of background RoIs identified). Increasing $\tau_b$ (percentage here) increases the fraction of RoIs treated as backgrounds. A small value of $\tau_b$ widens the search space for the combinatorial function when mining unknowns causing a small drop in performance due to confusions between true background and foreground unknowns. On the other hand, very large values of $\tau_b$ shrink the search space oftentimes considering unknown foregrounds as background objects showing a steep drop in performance.
>
> *Table R3: Ablation Experiments on the exclusion criterion $\tau_e$ and background budget $\tau_b$ in CROWD-D. The submodular function is kept constant as Graph-Cut in CROWD-D and a CROWD-FL based learning objective is chosen in CROWD-L under constant machine seed.*
>
> | **Method**             | Value    | **Task 1 U-Recall** | **Task 1 mAP** | **Task 2 U-Recall** | **Task 2 mAP** | **Task 3 U-Recall** | **Task 3 mAP** | **Task 4 mAP** |
> |------------------------|----------|---------------------|---------------|---------------------|---------------|---------------------|---------------|---------------|
> | **CROWD ($\tau_e$)**     | 0.05     | 57.5                | 61.7          | 52.7                | 47.1          | 68.2                | 41.5          | 38.0          |
> |                                | **0.2**  | 57.9                | 61.7          | 53.6                | 47.8          | 69.6                | 42.4          | 38.5          |
> | $\tau_b$ = 30%              | 0.5      | 53.3                | 59.5          | 49.4                | 45.9          | 65.0                | 40.1          | 37.8          |
> |------------------------|----------|---------------------|---------------|---------------------|---------------|---------------------|---------------|---------------|
> | **CROWD ($\tau_b$)**  | 10%      | 57.9                | 61.7          | 53.1                | 46.8          | 69.2                | 41.3          | 37.4          |
> |                               | **30%**  | 57.9                | 61.7          | 53.6                | 47.8          | 69.6                | 42.4          | 38.5          |
> | $\tau_e$ = 0.2                  | 50%      | 55.4                | 60.0          | 50.0                | 42.1          | 63.7                | 38.9          | 34.5          |
>
>
> We appreciate the reviewer’s comment on the societal impact of CROWD. We agree that object detection is a largely applied technique in real-world scenarios and OWOD helps object detectors continually adapt to new settings. We will include this in the questionnaire in the immediate next revision.

---

> > ### Author Response · Authors · 2025-08-04
> >
> > Respected Reviewer,
> >
> > We would like to thank you once again for your review. We hope our answers through the rebuttals have answered all your questions. Please let us know if you have any further questions/clarifications, so that we can further improve the quality of our paper.
> >
> > Thanking you,
> >
> > Authors

---

> > ### Comment · Reviewer_bAGd · 2025-08-04
> > **Comment on Rebuttal**
> >
> > I read the rebuttal from authors.
> >
> > My concerns regarding lack of study of impact of mentioned parameters have been addressed. I maintain the positive score.

---

### Official Review · Reviewer_3Pty · 2025-07-02

**Clarity:** 2
**Significance:** 3
**Originality:** 3
**Rating:** 4
**Confidence:** 4

**Summary:**

This paper addresses two major bottleneck challenges in Open World Object Detection, namely confusion between known and unknown objects, and catastrophic forgetting of previous learned classes. This paper proposes a novel framework called CROWD (Combinatorial Open World Detection), innovatively reformulating the OWOD problem as an interleaved task of combinatorial optimization-based data discovery (CROWD-Discover) and representation learning (CROWD-Learn). The paper utilizes the submodular conditional gain function to effectively mine representative unknown instances that are significantly different from known classes from unlabeled data. The paper employs different novel combinatorial objective functions to decouple the features of known and unknown classes while maintaining the discriminative consistency within known classes, providing new ideas for the above two key issues. Experimental results show that this method achieves remarkable results in standard OWOD benchmark tests, significantly improving the recall rate of unknown objects and the detection accuracy of known classes.

**Questions:**

1. There seems to be some ambiguity between the selection of Bt and Ut in the framework proposed in the article described in Figure 1 and the subsequent description.
2. Whether using the replay buffer to solve the problem of catastrophic forgetting can only alleviate it to a certain extent or completely solve it. If a large number of new categories are added, will the problem still occur due to the fixed size of the buffer?
3. Have any experiments been conducted to verify the method of combining the designed loss function with the replay buffer? How should it be set to achieve the best effect? What’s the best settings of the replay buffer?
4.  The paper selected a set of submodular function combinations with the best performance for the CROWD framework through experiments on the M-OWOD benchmark. However, it failed to verify whether this set of function combinations is also the optimal choice on other benchmarks with different data characteristics (such as S-OWOD), and no relevant experiments on the robustness of this combination were provided.

**Ethical Concerns:**

["NO or VERY MINOR ethics concerns only"]

**Final Justification:**

The response makes some clarification regarding the "replay buffer", and some Figures will be optimized in the further version . Overall, I'm positive about the article and maintain my rating for the current version.

**Limitations:**

Yes

**Quality:**

3

**Strengths And Weaknesses:**

Strengths
1. This paper is dedicated to solving two important problems encountered in open-world object detection and proposes a novel and interesting approach. The article has clear logic and sufficient innovation.
2. The comparative experiments in the article are very rich, which fully verify the feasibility of the framework proposed in the article.
3. The ablation experiments are rigorously designed, fully analyzing the contributions of each innovative module. The paper not only demonstrates the superiority of the overall solution but also verifies the necessity and effectiveness of each component through a detailed ablation study, making the sources of the method's advantages clear and the argumentation process very solid.

Weaknesses
1. The paper's experiments on the sensitivity of hyperparameters involved in the network structure are not enough sufficient. For example, the article mentions a small replay buffer, but does not conduct relevant experimental analysis on specific settings such as the buffer size and sampling strategy.
2. When comparing different SCG functions, although the paper selects a combination with relatively good results, it lacks a comparison of the performance of these SCG functions on different datasets,such as S-OWOD. The robustness of the model mentioned in the article still needs further relevant experiments to prove.

---

> ### Author Rebuttal · Authors · 2025-07-29
>
> **[W1] “The paper’s experiments …”** - We understand the concern of the reviewer regarding the “replay buffer” used in our experiments. We would like to clarify that the use of replay buffer is not unique to CROWD and has been used in several benchmarks like ORE (Joseph et al., 2021) and OrthogonalDet (Sun et al., 2024). Since we built our work on top of OrthogonalDet’s framework we used the exact same replay buffer which was publicly released (a fixed image list which is also same in ORE) in their codebase, thus did not require ablation. The number of unknown instances mined in CROWD-D however is a contribution of CROWD and has been ablated upon in Table 8 of the Appendix.
>
> **[W2] “When comparing different …”** - Since the end to end model training and inference takes roughly 32 hours in CROWD requiring a relatively large compute infrastructure we only conduct our ablation experiments on M-OWODB to limit compute and budgetary needs. This also follows the convention in SoTA approaches. However, we would like to point the reviewer to Table 2 (in main paper) where we demonstrate results on S-OWOD benchmark (as requested) for the exact same settings determined from the ablation study in Table 5 showing improvements in performance over SoTA approaches in ¾ tasks.
>
> **[Q1] “There seems …”** - We apologize for the confusion caused and would like to clarify that Fig. 1 was intended to be a very high-level overview of CROWD so some minor technical details have been omitted. Here, we provide a step by step analogy between Fig. 1 and our method description and will update the figure to show all technical details mentioned below -
>
> (1) At first we show that a model trained only on known classes has no knowledge of unknowns (discarded as low confidence BG by the NMS) (top subfigure in Fig.1).
>
> (2) We then pull out the RoI proposals before application of NMS to include the BG proposals and discard proposals with the lowest confidence as true background. This is not currently shown in Fig. 1 but we highlight this in Fig. 3.
>
> (3) Given the known class embeddings and the true backgrounds we apply CROWD-D to select the unknown class examples (middle subfigure in Fig. 1) and store them as $U^t$ for each task (also shown in Fig. 1).
>
> (4) Finally, we apply CROWD-L to learn both known and unknown classes in incremental finetuning fashion (bottom subfigure in Fig. 1).
>
> **[Q2] “Whether using …”** - We appreciate the question raised by the reviewer in this context. Using replay buffers (a common practice in literature both in OWOD and incremental learning) is a compute and memory efficient technique but does not completely solve catastrophic forgetting. According to our claims in CROWD we see that mining representative unknowns (CROWD-D) alongside a fixed buffer (this is predetermined from earlier research) and a better learning objective (CROWD-L) reduces the impact of forgetting. This is definitely a topic of continuing research and has been mentioned in our limitations as well.
>
> **[Q3] “Have any …”** - Continuing from our response to a similar question in [W1] above we would like to emphasize that the replay buffer used in CROWD is not determined by us but is a fixed list list of images for both M-OWOD and S-OWOD benchmarks released publicly in ORE, OrthogonalDet etc. This fixed imagelist has been used in most existing SoTA approaches during finetuning to maintain a fair comparison.
>
> **[Q4] “The paper selects …”** - We agree with the reviewer that we conduct our ablations only on the M-OWOD benchmark but use the exact same settings across tasks in both M-OWOD and S-OWOD benchmarks. From our literature review, this seems to be a common practice in literature adopted in SoTA approaches like PROB, OrthogonalDet, OW-DETR etc. Similar to our response in [W2] we would like to point the reviewer to Table 2 where we provide results on both M-OWOD and S-OWOD benchmarks keeping the exact same hyper-parameter settings as the ablation study.

---

> > ### Author Response · Authors · 2025-08-04
> >
> > Respected Reviewer,
> >
> > We would like to thank you once again for your review. We hope our answers through the rebuttals have answered all your questions. Please let us know if you have any further questions/clarifications, so that we can further improve the quality of our paper.
> >
> > Thanking you,
> >
> > Authors

---

> > ### Comment · Reviewer_3Pty · 2025-08-05
> >
> > Thanks for the author's response. I noticed the clarification regarding the "replay buffer" and the potential for further optimization of some Figures. Overall, I'm positive about the article and maintain my rating for the current version.

---

### Official Review · Reviewer_MYTT · 2025-07-03

**Clarity:** 2
**Significance:** 2
**Originality:** 2
**Rating:** 4
**Confidence:** 4

**Summary:**

The paper proposes a new method named CROWD (CombinatoRial Open-World Detection) for detecting known and unknown (semantic) object classes in 2D camera images. The method first trains an object detector in a supervision setting to detect known object classes. After that, the predictions of the trained detector are used to discover unknown objects in the “background”. The filtered background objects and known class instances are then used to train the detector again. The paper claims significantly improved recall for detecting unknown objects and also improves detection performance for known classes.

**Questions:**

1. Why is the selected orange bounding box for the chair in Figure 1 a different size than the red bounding box, which is part of the “background” bounding boxes. By just selecting a bounding box, the size should remain the same, right?

**Ethical Concerns:**

["NO or VERY MINOR ethics concerns only"]

**Final Justification:**

The authors are thanked for their rebuttal. My questions are answered. I am satisfied with the response. I changed my rating to borderline accept. If the authors incorporate the rebuttal in their paper, it can become solid work.

**Limitations:**

Yes.

**Paper Formatting Concerns:**

There are a couple of inconsistencies with respect to the NeurIPS conventions/template, namely:

1. In Figure 1, the is 1a and 1b, but it seems that 1a is just empty. The authors use the captions of 1a and 1b to explain a single figure which is out in 1b. It would be better to fuse all the captions into a single caption and put the caption below the figure. In addition, for some words in the figure, it is hard to determine to what image/part it belongs in the figure.
2. Put the captions of the subfigures of Figure 3 below the subfigures.
3. Split equation 1 into three different equations as three formulas are written in equation 1.
4. Put the names of the plots in Figure 4 as subcaptions below the plots.

**Quality:**

2

**Strengths And Weaknesses:**

Strengths
1. Paper claims to improve recall for detecting unknown object class instances by 2.4 and also improves detection performance for known object class instances.
2. The considered task is relevant in the current AI field, i.e. a label-efficiënt method is proposed as a object detector is trained to detect object classes that where not part of the training dataset labels.

Weaknesses
1. The first part of the title is very general “Looking Beyond the Known”. It basically states the same as “Open-World Object Detection” which is in the second part of the title. It would better to name the method name in the title or to put the characteristics of the proposed method in the title.
2. The method name CROWD is chosen a bit randomly, the “r” is just one of the letters of “CombinatoRial”.
3. Line 27 is unclear. It is unclear whether “previously learned classes” are about “classes that were unknown” or about “known classes”. In line 29, it seems that “unknown” classes were meant.
4. Lines 30-31 refer to the Wilderness Impact scores in Table 2 but these cannot be found in Table 2.
5. The symbols introduced in lines 35-36 give the impression of an exponent. Subscript would be better.
6. The second part of the introduction is mainly a methodology. Even some sections of the methodology are named. The last part of the introduction (58-75) is very vague / hard to understand. Last sentence of line 75 contains typo, it should be “Our main contributions are:”.
7. Three contributions are stated but the third one is just empirical support for the first two contributions (both are designed components for the proposed method).

---

> ### Author Rebuttal · Authors · 2025-07-29
>
> **[W1] “The first part …”** - We understand the concern of the reviewer on the title. We would like to clarify that the reasoning behind the use of “Looking Beyond the Known” was to highlight the importance of discovering high-quality unknown examples (which is achieved in CROWD-D). We would be happy to update the title with the technical details of our method in the immediate next revision.
>
> **[W2] “The method name …”** - The choice of CROWD as an abbreviation highlights that our model is able to identify potential unknown foreground objects within a “crowd” of background RoIs. In our defence, the use of intermittent letters like “r” seems to be a common practise in literature, even within the domain of OWOD eg. PROB (Zohar et al., 2023) which is abbreviated form of “**Pr**obabilistic **Ob**jectness”.
>
> **[W3] “Line 27 …”** - We thank the reviewer for pointing this out. The phrase pointed to refers to “known classes” in current task $T^t$. We will definitely update this in the paper for better readability. In line 29 “unknowns” do refer to unknown classes.
>
> **[W4] “Lines 30-31 … “** - We apologize for the oversight. Wilderness Impact and Absolute Open-Set Error (ASE) are demonstrated in Table 3 of the main paper. We will correct this in the immediate next revision.
>
> **[W5] “The symbols …”** - We understand the concern of the reviewer and would be happy to replace the current symbols with the “t” in subscript. However, in our defence, these symbols of known and unknown classes are used as in OrthogonalDet (Sun et al., 2024) who use $t$ in superscript.
>
> **[W6] “The second part …”** - We appreciate the criticism of the reviewer on this. Our goal here was to give a complete technical overview of our CROWD approach in a summarized form under the introduction with suitable references to details from the methodology section. We will be happy to dial down the amount of details and provide a high-level overview in the immediate next revision.
>
> **[W7] “Three contributions …”** - We agree that the third claim in our paper is an empirical validation of the previous two claims. We would also like to emphasize that the empirical prowess of methods in OWOD has been enlisted as a core contribution in several previous SoTA methods like ORE (Joseph et al., 2021), PROB (Zohar et al., 2023), OW-DETR (Gupta et al., 2022) and OrthogonalDet (Sun et al., 2024). We follow a similar pattern in writing our contributions. These empirical gains (even though are results of the method used) show the applicability of CROWD to general Open-World scenarios which we deem to be a critical contribution.
>
> Please find below our responses to the questions raised -
>
> **[Q1] “Why is the …”** - We understand the concern of the reviewer and would like to emphasize that Figure 1 is a broad overview of the entire learning mechanism in CROWD and the instances shown are for illustration purposes only. The number of RoIs returned by the RPN for each image in our model is set to 500 (lines 263-264) with a large number of overlapping ones (since NMS is not applied in this stage) which we could not show in the figure (on the left). The figure on the right (with unknown instances plotted) highlights only the selected RoIs (3 / 10 selected) due to which the sizes appear different. We do acknowledge the difference in sizes and will ensure they are consistent in the immediate next revision.
>
> Please find below our responses to the formatting issues raised in the review -
>
> **[F1] “In Figure 1 …”** - We thank the reviewer for the suggestion and will update the figure as suggested in the next revision. We clarify here that Fig. 1a and 1b does not refer to two different sub-figures and points to the complete process of selection and learning in CROWD. The top two rows of Fig.1 resembles CROWD-D denoted as 1a and the bottom row points to CROWD-L, denoted as 1b.
>
> **[F2] “Put the …”** - We thank the reviewer and will update the position of the subfigure titles.
>
> **[F3] “Split the …”** - We understand the confusion of the reviewer. We will keep only the final formulation of $L_{\text{CROWD}^{\text{cross}}}$ here and split the equation to accommodate for space. The remaining would be referenced from the Appendix where we provide the detailed derivations.
>
> **[F4] “Put the …”** - We thank the reviewer and will update the position of the subplot titles.

---

> > ### Author Response · Authors · 2025-08-04
> >
> > Respected Reviewer,
> >
> > We would like to thank you once again for your review. We hope our answers through the rebuttals have answered all your questions. Please let us know if you have any further questions/clarifications, so that we can further improve the quality of our paper.
> >
> > Thanking you,
> >
> > Authors

---

> ### Comment · Area_Chair_rkGA · 2025-08-08
>
> Dear Reviewer MYTT,
>
> As the Author-Reviewer discussions will end, please read the author's response and engage in the discussion with the authors. Are the authors' answers satisfactory for your questions? If so, would you modify your score accordingly? If not, could you leave the reasons?
>
> Thanks and regards,
> Area Chair

---

### Official Review · Reviewer_9jJR · 2025-07-03

**Clarity:** 3
**Significance:** 3
**Originality:** 2
**Rating:** 5
**Confidence:** 3

**Summary:**

This paper proposes a novel framework for Open-World Object Detection (OWOD) named Combinatorial Open-World Detection (CROWD), which consists of two tightly coupled components: CROWD-Discover for targeted unknown instance mining, and CROWD-Learn for submodular objective-based representation learning. Unlike previous approaches that rely on objectness heuristics or clustering, CROWD formalizes OWOD as a combinatorial problem using submodular conditional gain (SCG) and submodular information measures (SIM). This formulation aims to both identify informative unknown instances and ensure robust feature disentanglement, thereby mitigating known-unknown confusion and catastrophic forgetting. The method achieves state-of-the-art performance across multiple OWOD benchmarks.

**Questions:**

1. Include runtime/memory benchmarks or FLOPs to better quantify the computational trade-offs of CROWD compared to baselines.
2. Provide a sensitivity analysis for thresholds τₑ and τ_b (e.g., a heatmap or performance curve).
3. Add qualitative failure cases where CROWD-D misidentifies background as unknown, to better understand its limitations in complex scenes.
2. In Figure 2, superscripts and subscripts are not very clear. Could u further explain them?

**Ethical Concerns:**

["NO or VERY MINOR ethics concerns only"]

**Final Justification:**

The response from authors addressed most of my concerns, so I'd like to increase my final rating from 4 to 5.

**Limitations:**

Yes

**Paper Formatting Concerns:**

null

**Quality:**

3

**Strengths And Weaknesses:**

### Strengths:
1. The paper articulates the inherent limitations of existing OWOD approaches, particularly in semantic confusion and forgetting, and motivates the need for a more principled and theoretically grounded framework. Modeling OWOD as a combinatorial data-discovery and learning task is a novel and theoretically sound perspective. The use of submodular functions introduces strong guarantees and interpretability.
2. The method demonstrates significant gains in unknown recall and competitive improvements in known-class accuracy on both M-OWOD and S-OWOD benchmarks, validating its effectiveness.
3. The paper is well-organized, with a clear separation of components (CROWD-D and CROWD-L), detailed methodology, and informative figures and tables that aid understanding.

### Weaknesses:
1. Lines 191–194 describe heuristics (e.g., filtering by objectness score and subset selection) to reduce computational cost. However, the actual computational complexity or runtime comparison to baselines is not discussed. Since CROWD introduces combinatorial subset selection, this may impact scalability. Could the authors provide complexity analysis or empirical runtime comparisons?
2. The framework heavily relies on two thresholds, τₑ (objectness threshold) and τ_b (background selection budget). Although empirically set (0.2 and 30%), the sensitivity of performance to these hyperparameters is not systematically analyzed. How robust is the system to variations in τₑ and τ_b? Could performance degrade significantly if these are misconfigured?
3. The paper positions CROWD-D as a more principled alternative to heuristic pseudo-labeling (e.g., ORE, PROB), yet concrete distinctions remain somewhat abstract. What is the core theoretical or empirical advantage of CROWD-D over attention-based or objectness-based mining (e.g., OW-DETR, CAT)? In real-world settings where unknowns and backgrounds may share features, how does CROWD-D avoid misidentifying complex background as unknown?
4. As shown in Table 5, the standalone gain from CROWD-L is relatively modest (~0.4–1.2% mAP over baseline), especially considering the additional modeling complexity. Could the authors elaborate on why the representation learning stage has limited standalone impact? Is it because CROWD-L primarily complements CROWD-D, or is the performance bounded by the quality of mined unknowns?

---

> ### Author Rebuttal · Authors · 2025-07-29
>
> **[W1] “Lines 191-194 …”** - We thank the reviewer for the detailed question. The reduction in computation cost referred to in lines 191-194 results from the exclusion of very low confidence RoI proposals potentially containing (line 2 in Algorithm 1). To put things numerically we contrast against existing SoTA approach OrthogonalDet (Sun et al., 2024) which models interactions between every possible pair of RoI features to calculate a relevance score (to select unknowns). For the Faster-RCNN family of object detectors the number of RoIs are 500 (as set in our work as well as OrthogonalDet) resulting in a 500 x 500 sized similarity kernel for every image in the dataset. On the other hand, excluding very low confidence proposals (thresholded on the objectness scores) in CROWD naturally reduces the number of RoIs to ~ 200-300 proposals reducing the compute overhead. Also, in terms of runtime we observe that on average OrthogonalDet requires about 36 hours to train, reweight and finetune on a particular task while our method CROWD can perform this in ~32 hours (using the same compute infrastructure and seed setting). This can be attributed to the joint impact of the combinatorial selection and faster convergence of the combinatorial objectives (similar results observed in SMILe (Majee et al., 2024) ) due to modeling of the learning task as a set problem (30k steps in OrthogonalDet per task vs. 24k steps in CROWD per task). We will release the log files from the best performing runs from our framework (built upon detectron2) upon acceptance showing this statistic.
>
> **[W2] “The framework …”** - We appreciate the reviewer’s constructive criticism on the hyper-parameters in CROWD. We clarify that $\tau_e$ is an exclusion threshold which reduces the search space of CROWD-D by eliminating RoIs which have a low confidence threshold. As shown in Tab. R1 below, increasing $\tau_e$ from 0 to 1 increases performance until $\tau_e = 0.2$ and then reduces. A lower value of $\tau_e$ allows for a large search space but includes a lot of noisy background objects leading to reduced selection performance. On the other hand a large value of $\tau_e$ can potentially earmark unknown foregrounds as background resulting in reduced performance.
>
> Keeping $\tau_e$ fixed at 0.2 we ablate $\tau_b$ which controls the selection budget for backgrounds (higher the value more are the number of background RoIs identified). Increasing $\tau_b$ (percentage here) increases the fraction of RoIs treated as backgrounds. A small value of $\tau_b$ widens the search space for the combinatorial function when mining unknowns causing a small drop in performance due to confusions between true background and foreground unknowns. On the other hand, very large values of $\tau_b$ shrink the search space oftentimes considering unknown foregrounds as background objects showing a steep drop in performance.
>
> *Table R1: Ablation Experiments on the exclusion criterion $\tau_e$ and background budget $\tau_b$ in CROWD-D. The submodular function is kept constant as Graph-Cut in CROWD-D and a CROWD-FL based learning objective is chosen in CROWD-L under constant machine seed.*
>
> | **Method**             | Value    | **Task 1 U-Recall** | **Task 1 mAP** | **Task 2 U-Recall** | **Task 2 mAP** | **Task 3 U-Recall** | **Task 3 mAP** | **Task 4 mAP** |
> |------------------------|----------|---------------------|---------------|---------------------|---------------|---------------------|---------------|---------------|
> | **CROWD ($\tau_e$)**     | 0.05     | 57.5                | 61.7          | 52.7                | 47.1          | 68.2                | 41.5          | 38.0          |
> |                                | **0.2**  | 57.9                | 61.7          | 53.6                | 47.8          | 69.6                | 42.4          | 38.5          |
> | $\tau_b$ = 30%              | 0.5      | 53.3                | 59.5          | 49.4                | 45.9          | 65.0                | 40.1          | 37.8          |
> |------------------------|----------|---------------------|---------------|---------------------|---------------|---------------------|---------------|---------------|
> | **CROWD ($\tau_b$)**  | 10%      | 57.9                | 61.7          | 53.1                | 46.8          | 69.2                | 41.3          | 37.4          |
> |                               | **30%**  | 57.9                | 61.7          | 53.6                | 47.8          | 69.6                | 42.4          | 38.5          |
> | $\tau_e$ = 0.2                  | 50%      | 55.4                | 60.0          | 50.0                | 42.1          | 63.7                | 38.9          | 34.5          |
>
>
> **[W3] “The paper positions …”** - We thank the reviewer for the insightful question. As noted in lines 90–92 of the main paper, CAT employs a dual pseudo-labeling strategy: a model-driven approach leveraging attention scores and confidence to mine unknowns, and an input-driven approach using selective search to propose unknown candidate regions independent of model predictions. Similarly, OW-DETR uses unmatched object queries with high backbone activation scores as unknowns during training.
>
> In contrast, CROWD-D takes a fundamentally different, principled approach by framing unknown mining as a greedy submodular maximization problem based on information-theoretic submodular conditional gain (SCG) functions (Sec. 3.3). Instead of relying solely on heuristic attention or objectness scores, CROWD-D first filters out clear background regions via low objectness scores (inspired by CAT) and then selects unknown candidates by maximizing the information gain relative to a fixed set of known RoIs. The optimization process also highlighted in Algorithm 2 (appendix) inherently encourages selection of examples that are maximally informative (high marginal gains) thereby dissimilar from known classes, reducing confusion with complex backgrounds. It is interesting to note that our selection strategy is further backed by strong theoretical guarantees (Khargonkar etal., 2021) and follows the laws of diminishing marginal return to achieve this behavior. Nonetheless, the effectiveness of CROWD-D does depend on the quality of underlying feature representations, motivating the joint use of CROWD-L to improve feature discriminability between known and unknown objects.
>
> **[W4] As shown in Table 5 …”** - We agree with the reviewer that the standalone gain of CROWD-L is 0.4 to 1.2% over SoTA approaches while CROWD-D shows up to 2$\times$ gain in unknown class recall. The selection process in CROWD-D can potentially inject some noisy pseudo-labels in the learning process as shown by the drop in performance from 47.0 to 45.2 in Table 5 (row 2 vs. row 4). To maintain the performance on known classes CROWD-L complements CROWD-D and enforces sufficient separation between known and unknown RoI features without sacrificing on the boost in unknown recall resulting from CROWD-D (last row in Table 5).
>
> **[Q1] “Include runtime …”** - We would like to request the reviewer to kindly refer to our comment [W1] in the same rebuttal where we describe the runtime costs incurred in CROWD over SoTA approach OrthogonalDet (Sun et al., 2024).
>
> **[Q2] “Provide a sensitivity  …”** - We would like to point the reviewer to our comment in [W2] of the rebuttal where we provide ablation experiments to show the impact of these hyperparameters.
>
> **[Q3] “Add qualitative …”** - We understand the concern of the reviewer and we will definitely include some qualitative failure cases in the Appendix of our paper in the immediate next revision.
>
> **[Q4] “In Figure 2 …”** - We understand the confusion of the reviewer regarding the symbols used in Fig. 2 (main paper). At first, we would like to point the reviewer to Table 6 of the Appendix where a glossary of symbols and their respective meanings in the context of CROWD has been provided. Particularly for Figure 2 we provide below a step by step explanation below -
>
>
> (1) Fig. 2 is a detailed overview of a single task within OWOD. A task $T^t$ with task identifier $t$ accepts a model $h^t$ and a replay buffer $\hat{K}^{t-1}$ containing examples from previously known objects.
>
> (2) At first we train $h^t$ on only currently known objects $K^t$ a replay buffer from which is stored as $\hat{K}^t$. For every example in $\hat{K}^t$ we also extract background RoIs $B^t$.
>
> (3) CROWD-D mines a set of unknowns $\hat{U}^t$ from $\hat{K}^t$ and $B^t$ which is used by CROWD-L.
>
> (4) CROWD-L learns an updated model $h^{t+1}$ by finetuning on $\hat{K}^{t-1}$ (previously known), $\hat{K}^t$ (currently known) and $\hat{U}^t$ (mined unknowns) using our novel learning objectives.
>
> (5) A replay buffer of the currently learnt objects $\hat{K}^t$ and the model weights $h^{t+1}$ are passed on to the next task.

---

> > ### Author Response · Authors · 2025-08-04
> >
> > Respected Reviewer,
> >
> > We would like to thank you once again for your review. We hope our answers through the rebuttals have answered all your questions. Please let us know if you have any further questions/clarifications, so that we can further improve the quality of our paper.
> >
> > Thanking you,
> >
> > Authors

---

> > > ### Author Response · Authors · 2025-08-08
> > > **Gentle reminder - less than 24 hours left in the discussion period**
> > >
> > > Respected Reviewer #9jJR,
> > >
> > > Thank you for your detailed review encouraging the findings of our approach alongside some very engaging questions. We sincerely hope that our responses have addressed all your questions adequately.
> > >
> > > This is a kind reminder that the rebuttal period will close in less than 24 hours. If you have any further questions or require clarifications regarding our paper, please feel free to reach out. We appreciate your time and consideration.
> > >
> > > Thank you,
> > >
> > >  Authors

---

> > ### Comment · Reviewer_9jJR · 2025-08-09
> > **Official comments by reviewer 9jJR**
> >
> > I appreciate the author’s detailed response, which addressed most of my issues and concerns. The step-by-step illustration in Figure 2 greatly helped me understand the proposed method. However, I believe the clarity of Figure 2 could be further improved in the next revision. Finally, I would like to increase my final rating from 4 to 5.

---

### Decision · Program_Chairs · 2025-09-17

**Decision:**

Accept (poster)

**Comment:**

This paper tackles the challenges of semantic confusion and catastrophic forgetting in the problem of open-world object detection. It introduces the CROWD framework, which combines combinatorial data discovery and representation learning. The former selects representative unknowns distinct from known classes, while the latter employs combinatorial objectives to disentangle representations and preserve discriminative power. Experiments show notable gains from the framework.

This paper received comments from five reviewers. The main concerns before the rebuttal are reflected in (1) the threshold involved in the framework; (2) some empirical performance gains are marginal; (3) algorithm details and implementation are not clear enough; (4) the ablation studies are not sufficient to understand the key contributions of the work. The authors provided detailed feedback, and all reviewers reached a consensus.

The AC has carefully reviewed the paper and the reviewers’ comments, and considers this to be a solid and meaningful work. The authors are encouraged to incorporate the reviewers’ constructive feedback in the final version to further strengthen the clarity and impact of this work.